# CAN LLMS REASON SOUNDLY IN LAW? AUDITING INFERENCE PATTERNS FOR LEGAL JUDGMENT

**Lu Chen**[1]   **Yuxuan Huang**[1,3]   **Yixing Li**[2]   **Dongrui Liu**[3]   **Qihan Ren**[1]   **Shuai Zhao**[1]
**Kun Kuang**[4]   **Zilong Zheng**[5]   **Quanshi Zhang**[1]*

[1]Shanghai Jiao Tong University
[2]The Chinese University of Hong Kong
[3]Shanghai Artificial Intelligence Laboratory
[4]Zhejiang University
[5]Beijing Institute for General Artificial Intelligence
`{lu.chen,renqihan,shuaizhao,zqs1022}@sjtu.edu.cn,`
`huangyuxuan@pjlab.org.cn,  yixingli@link.cuhk.edu.hk,`
`liudongrui@pjlab.org.cn,  kunkuang@zju.edu.cn,`
`zilongzheng0318@ucla.edu`

## ABSTRACT

This paper presents a method to analyze the inference patterns used by Large Language Models (LLMs) for judgment in a case study on legal LLMs, so as to identify potential incorrect representations of the LLM, according to human domain knowledge. Unlike traditional evaluations on language generation results, we propose to evaluate the correctness of the detailed inference patterns of an LLM behind its seemingly correct outputs. To this end, we quantify the interactions between input phrases used by the LLM as primitive inference patterns, because recent theoretical achievements (26; 42) have proven several mathematical guarantees of the faithfulness of the interaction-based explanation. We design a set of metrics to evaluate the detailed inference patterns of LLMs. Experiments show that even when the language generation results appear correct, a significant portion of the inference patterns used by the LLM for the legal judgment may represent misleading or irrelevant logic[1].

## 1 INTRODUCTION

Large language models (LLMs) (57; 29; 1; 15; 24) have demonstrated state-of-the-art performance on a wide range of tasks. However, for high-stakes applications, only high accuracy of the generated outputs is still insufficient to ensure the reliability of LLMs (37; 59; 23; 53; 51) for the following main reasons. (1) We find that even when a top-tier LLM generates correct tokens, the LLM still relies on problematic *inference patterns* to generate the next token. (2) In particular, for LLMs towards legal judgment (16; 13; 8; 4; 35), such problematic inference patterns directly influence the choice of the LLM among multiple seemingly acceptable judgments, which constitutes an encroachment upon domains traditionally recognized as within judges' discretionary authority. Thus, this would introduce significant potential unfairness and risks.

Therefore, in this paper, we focus on LLMs for legal judgment as a case study, as it serves as a typical high-stakes application. We explore the intense problematic inference patterns used by an LLM for judgments, and discuss the potential harm of these inference patterns. In fact, the first problem in this study is whether an LLM's prediction can be faithfully decomposed into a set of inference patterns. Recent works in explainable AI (47; 50; 42; 26; 39; 6) have demonstrated that **the inference score of a deep network can be faithfully represented by a set of interactions between input features**. As shown in Figure 1, an *interaction* extracted from an LLM captures a nonlinear relationship between

---

*Quanshi Zhang is the corresponding author. He is with the Department of Computer Science and Engineering, the John Hopcroft Center, at the Shanghai Jiao Tong University, China.

[1]The names used in the legal cases follow an alphabetical convention, *e.g.*, Andy, Bob, Charlie, etc., which do not represent any bias against actual individuals.

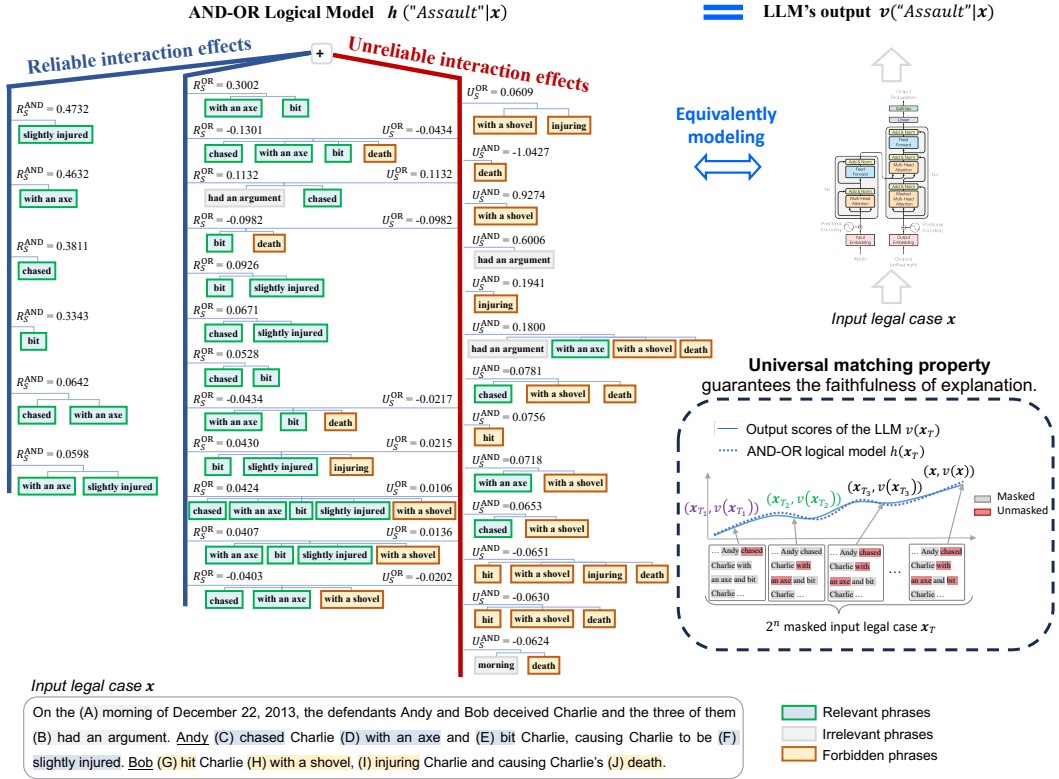

Figure 1: Correctness of the detailed inference patterns of an LLM. The AND-OR logical model $h(\cdot)$ accurately fits the output score of the LLM $v(\cdot)$ when making the judgment "***Assault***" for Andy, $h(\text{"}Assault\text{"}|\mathbf{x}) = v(\text{"}Assault\text{"}|\mathbf{x})$, no matter how the input legal case $\mathbf{x}$ is masked in the bottom-right figure. black edges connect *reliable interaction effects* ($R_S^{\text{AND}}$ and $R_S^{\text{OR}}$) that contribute to the output score $v(\text{"}Assault\text{"}|\mathbf{x})$, typically aligning with legal domain knowledge. Red edges connect *unreliable interaction effects* ($U_S^{\text{AND}}$ and $U_S^{\text{OR}}$) that contribute to $v(\text{"}Assault\text{"}|\mathbf{x})$, often reflecting problematic patterns used by the LLM for the judgment.

input tokens, and contributes a numerical score that quantifies their joint influence on the LLM's prediction.

Despite above theoretical achievements, in this paper, we focus on a crucial yet long-overlooked issue in the community, *i.e.*, the correctness of detailed representations of an LLM. It is still unknown *(1) how many problematic interactions are modeled in LLMs (e.g., legal LLMs), and (2) to what extent these interactions influence legal judgments.*

In particular, we obtain three findings. **(1)** We find that over half of interactions actually represent clearly unreasonable or even incorrect justifications for the judgment predictions. **(2)** Although the appearance of long-chain reasoning capabilities exhibited in chains-of-thought prompting (55; 29; 57; 37), we find that the essence is simple interactions of *local* tokens to guess judgments, even just like a bag-of-the-words model (36). **(3)** We find that LLMs tend to model a large number of canceling interactions, where positive and negative contributions between input tokens offset each other, which often represent unreliable noise patterns.

**Risk warning or Benchmark.** We acknowledge that we cannot exhaustively analyze all legal cases[2], but **our objective is to** provide sufficient examples to alert the deep learning community to the severity of representational flaws reflected by interactions encoded in LLMs. It is because our experiments show that the representation quality of current LLMs fails far short of supporting the benchmark evaluation of detailed interaction patterns. For example, Figure 1 shows LLMs use a large

---

[2]In addition to the large number of legal provisions, the variation in laws across countries presents another challenge.

number of problematic interactions to make judgments, making it hard to compare the quality of interactions in different LLMs in a meaningful way.

Instead, we choose to illustrate a wide range of problematic interaction patterns. While not exhaustive[2], in this paper, let us simply introduce three common types of potential representational flaws frequently observed in LLMs: **(1)** LLMs tend to make judgments based on semantically irrelevant phrases; **(2)** LLMs often make judgments using the behavior of incorrect entities; **(3)** LLMs tend to produce judgments that are biased by identity discrimination.

Experiments have shown that even when the LLM generated correct target tokens, a significant portion of the interactions encoded by the LLM for the legal judgment are unreliable. This reflects a significant yet long overlooked problem for LLMs.

## 2 EVALUATING DETAILED INFERENCE PATTERNS USED BY LLMS

### 2.1 PRELIMINARIES: EXTRACTING INTERACTIONS AS THEORETICALLY GUARANTEED INFERENCE PATTERNS

Recent advancements in explanation theory (26; 39; 42; 6) have proven that using an AND-OR logical model can accurately match all varying outputs of an LLM on exponentially augmented inputs. Specifically, given an input prompt $\mathbf{x} = [x_1, x_2, \cdots, x_n]^\mathsf{T}$ with $n$ input phrases indexed by $N = \{1, 2, ..., n\}$, where each input phrase represents a semantic unit, such as a token, a word, or a phrase/short sentence. Then, let $v(\mathbf{x}) \in \mathbb{R}$ denote the *scalar* output score of generating a sequence of the target $m$ tokens $[y_1, y_2, \cdots, y_m]$, as follows.

$$v(\mathbf{x}) \stackrel{\text{def}}{=} \sum_{t=1}^{m} \log \frac{p(y = y_t | \mathbf{x}, \mathbf{Y}_t^{\text{previous}})}{1 - p(y = y_t | \mathbf{x}, \mathbf{Y}_t^{\text{previous}})} \tag{1}$$

where $\mathbf{Y}_t^{\text{previous}} \stackrel{\text{def}}{=} [y_1, y_2, \cdots, y_{t-1}]^\mathsf{T}$ represents the sequence of the previous $(t-1)$ tokens before generating the $t$-th token. $p(y = y_t | \mathbf{x}, \mathbf{Y}_t^{\text{previous}})$ denotes the probability of generating the $t$-th token. In particular, $\mathbf{Y}_1^{\text{previous}} = []$.

Theorem 1 proves that given an input prompt $\mathbf{x}$, the output score of the LLM $v(\mathbf{x})$ can be well predicted/fitted by the following AND-OR logical model $h(\mathbf{x})$, no matter how we enumerate all $2^n$ masked states of the input prompt[3].

$$h(\mathbf{x}_{\text{mask}}) \stackrel{\text{def}}{=} h(\mathbf{b}) + \sum_{S \in \Omega^{\text{AND}}} \mathbb{1}_{\text{AND}}(S|\mathbf{x}_{\text{mask}}) \cdot I_S^{\text{AND}} + \sum_{S \in \Omega^{\text{OR}}} \mathbb{1}_{\text{OR}}(S|\mathbf{x}_{\text{mask}}) \cdot I_S^{\text{OR}} \tag{2}$$

• **The AND trigger function** $\mathbb{1}_{\text{AND}}(S|\mathbf{x}_{\text{mask}}) \in \{0, 1\}$ represents a binary AND logic (also termed an *AND interaction pattern*) between input phrases of the masked sample $\mathbf{x}_{\text{mask}}$ in $S$. It returns 1 if **all** phrases in $S$ are present (not masked) in $\mathbf{x}_{\text{mask}}$; otherwise, it returns 0. $I_S^{\text{AND}}$ is the scalar weight. Here, $\Omega^{\text{AND}} \subseteq 2^N = \{S \subseteq N\}$ represents the set of AND interaction patterns. $\mathbf{b}$ is a scalar bias.

• **The OR trigger function** $\mathbb{1}_{\text{OR}}(S|\mathbf{x}_{\text{mask}}) \in \{0, 1\}$ represents a binary OR logic (also termed an *OR interaction pattern*) between input phrases of the masked sample $\mathbf{x}_{\text{mask}}$ in $S$. It returns 1 when **any** phrase in $S$ appears (not masked) in $\mathbf{x}_{\text{mask}}$; otherwise, it returns 0. $I_S^{\text{OR}}$ is the scalar weight. Here, $\Omega^{\text{OR}} \subseteq 2^N = \{S \subseteq N\}$ denotes the set of OR interaction patterns.

**Theorem 1** (Universal matching property, proof in Section C)**.** *When scalar weights in the logical model are set to* $\forall S \subseteq N, I_S^{\text{AND}} \stackrel{\text{def}}{=} \sum_{T \subseteq S} (-1)^{|S|-|T|} v_{and}(\mathbf{x}_T)^4$ *and* $I_S^{\text{OR}} \stackrel{\text{def}}{=}$

---

[3]We followed (26) to obtain two discrete states for each input phrase, *i.e.*, the masked and unmasked states. Therefore, given an input prompt with $n$ phrases, there are $2^n$ possible masked states of the input prompt. To obtain the masked sample $\mathbf{x}_T$, we replaced the embedding of each token in the input phrase $i \in N \setminus T$ with the baseline value $b_i \in \mathbb{R}^d$, where $d$ is the embedding dimension of each token. The baseline value $b_i$ was trained as described in (40). Please see Section L.6 for details.

[4]The numerical effect of AND interaction pattern $I_S^{\text{AND}}$ is also known as the Harsanyi dividend (18) in the cooperative game theory.

$-\sum_{T \subseteq S}(-1)^{|S|-|T|}v_{or}(\mathbf{x}_{N \setminus T})$, *subject to* $v_{and}(\mathbf{x}_T) + v_{or}(\mathbf{x}_T) = v(\mathbf{x}_T)$, $\mathbf{b} = v(\mathbf{x}_\emptyset)$, *then we have*

$$\forall T \subseteq N, v(\mathbf{x}_T) = h(\mathbf{x}_T) \tag{3}$$

*where* $\mathbf{x}_T$ *is the masked sample[3] that each input variable* $i \in N \setminus T$ *is masked.* $v(\mathbf{x}_T)$ *is the LLM's scalar output score of the masked sample* $\mathbf{x}_T$[3]. $\Omega^{\text{AND}} = 2^N = \{S \subseteq N\}$, $\Omega^{\text{OR}} = 2^N = \{S \subseteq N\}$.

Theorem 1 shows that an AND-OR logical model $h(\cdot)$ in Equation (2) can well predict/match all output score of the LLM $v(\cdot)$ on all $2^n$ enumerated masked states[3] of the input prompt $\mathbf{x}$. **It partially guarantees that we can roughly consider each AND-OR interaction logic in the logical model $h(\cdot)$ represents an AND-OR inference pattern equivalently used by the LLM.**

**Sparsity of interaction patterns (inference patterns) and settings of $\Omega^{\text{AND}}$ and $\Omega^{\text{OR}}$.** Another issue is the conciseness of explanation. To this end, Ren et al. (42) have proven that the logical model obtained in Theorem 1 can be compressed into a concise AND-OR logical model by pruning all interactions with almost zero weight $I_S^{\text{AND}}$ and $I_S^{\text{OR}}$. Specifically, given an input prompt $\mathbf{x}$ with $n$ input phrases, there are only $\mathcal{O}(n^p)$ interaction patterns have considerable numerical scores. All other interactions have negligible numerical scores, *i.e.*, $I_S^{\text{AND}}, I_S^{\text{OR}} \approx 0$. It is usually found $1.5 \le p \le 2.0$. This guarantees a deep network to be explained concisely.

**Interaction extraction (pseudo-code in Algorithm 1).** For implementation, the concise AND-OR logical model can be obtained by setting $v_{\text{and}}(\mathbf{x}_T) = 0.5v(\mathbf{x}_T) + \gamma_T$ and $v_{\text{or}}(\mathbf{x}_T) = 0.5v(\mathbf{x}_T) - \gamma_T$ in Theorem 1, with a set of learnable parameters $\{\gamma_T | T \subseteq N\}$. We follow (61) to learn the parameters $\{\gamma_T | T \subseteq N\}$, and extract the sparest (the simplest) AND-OR interaction explanation using a LASSO-like loss function, *i.e.*, $\min_{\{\gamma_T\}} \sum_{S \subseteq N, S \ne \emptyset}[|I_S^{\text{AND}}| + |I_S^{\text{OR}}|]$. All salient interactions in $\Omega^{\text{AND}} \overset{\text{def}}{=} \{S \subseteq N : |I_S^{\text{AND}}| > \tau\}$ and $\Omega^{\text{OR}} \overset{\text{def}}{=} \{S \subseteq N : |I_S^{\text{OR}}| > \tau\}$ are selected to construct the logical model for explanation, where $\tau$ is a small threshold.

## 2.2 RELEVANT PHRASES, IRRELEVANT PHRASES, AND FORBIDDEN PHRASES

In this subsection, we annotate the *relevant, irrelevant,* and *forbidden* phrases in the input legal case, in order to accurately identify the reliable and unreliable interaction effects used by the legal LLMs for the legal judgment (see Figure 1). Here, an input phrase can be set as a token, a word, or a phrase.

Specifically, we engage 16 legal experts and volunteers[5] to manually partition the set of all input phrases $N$ into three mutually disjoint subsets, *i.e.*, the set of relevant phrases $\mathcal{R}$, the set of irrelevant phrases $\mathcal{I}$, and the set of forbidden phrases $\mathcal{F}$, subject to $\mathcal{R} \cup \mathcal{I} \cup \mathcal{F} = N$, with $\mathcal{R} \cap \mathcal{I} = \emptyset$, $\mathcal{R} \cap \mathcal{F} = \emptyset$, and $\mathcal{I} \cap \mathcal{F} = \emptyset$, according to their legal domain knowledge, as follows.

**Phrase annotation.** We first clarify principles to guide legal experts to annotate different types of phrases for judgments according to their legal domain knowledge.

(1) Generally speaking, **relevant phrases** refer to phrases that are closely related to, or directly contribute to the legal judgment result, based on their ground-truth relevance to the judgment result. For example, as Figure 1 shows, there are 10 informative input phrases chosen by legal experts. Among them, $\mathcal{R} = \{[chased], [with an axe], [bit], [slightly injured]\}$ are the direct reason for the judgment "*Assault*" for Andy, thereby being annotated as relevant phrases. In the computation of interactions, all tokens in the brackets [] are taken as a single input phrase.

(2) **Irrelevant phrases** are phrases that describe the defendant but are not sensitive phrases that directly contribute to the judgment result. For example, as Figure 1 shows, $\mathcal{I} = \{[morning], [had an argument]\}$ are *not* the direct reason for the judgment "*Assault*" for Andy, thereby being annotated as irrelevant phrases.

(3) **Forbidden phrases** are usually sensitive yet misleading phrases in the legal case, *e.g.*, phrases describing incorrect defendant. For example, as Figure 1 shows, $\mathcal{F} = \{[hit], [with a shovel], [injuring], [death]\}$ should not influence the judgment for Andy, because these words describe the actions and consequences of Bob, not actions of Andy, thereby being annotated as forbidden phrases for Andy.

---

[5]In particular, there are two senior legal experts who have been practicing for over 10 years.

Please see Section I for more examples of the annotated relevant phrases, irrelevant phrases, and forbidden phrases in real legal cases.

In particular, we set up **two principles** to guide 16 legal experts and volunteers to annotate phrases to enable a convincing evaluation. Please see Section I for detailed principles. We acknowledge that the above three types of phrases are not a complete enumeration of all problematic phrases in legal cases. Instead, this paper just aims to illustrate the existence of a large ratio of unreliable inference patterns used by the LLMs, rather than exhausting all potential issues with an LLM.

## 2.3 RELIABLE AND UNRELIABLE INTERACTION EFFECTS

Since the scalar weight $I_S^{\text{AND}}$ (or $I_S^{\text{OR}}$) denotes the numerical effect for the interaction (or called *interaction effect* for short), the annotation of *relevant, irrelevant,* and *forbidden* phrases enables us to decompose the overall interaction effects $I_S^{\text{AND}}$ and $I_S^{\text{OR}}$ in Theorem 1 into reliable effects ($R_S^{\text{AND}}$ and $R_S^{\text{OR}}$) and unreliable effects ($U_S^{\text{AND}}$ and $U_S^{\text{OR}}$), *i.e.,* $I_S^{\text{AND}} \overset{\text{decompose}}{=} R_S^{\text{AND}} + U_S^{\text{AND}}$ and $I_S^{\text{OR}} \overset{\text{decompose}}{=} R_S^{\text{OR}} + U_S^{\text{OR}}$. The absolute effect ($|I_S^{\text{AND}}|$ and $|I_S^{\text{OR}}|$) is termed the *interaction strength*.

In this way, we can define **reliable interaction effects** ($R_S^{\text{AND}}$ and $R_S^{\text{OR}}$) as interaction effects that align with human domain knowledge, and usually contain relevant phrases and exclude forbidden phrases. In contrast, **unreliable interaction effects** ($U_S^{\text{AND}}$ and $U_S^{\text{OR}}$) are defined as interaction effects that do not match human domain knowledge, which are attributed to irrelevant or forbidden phrases.

**Reliable interactions and unreliable interactions.** Figure 1 further provides an example of using AND-OR interactions to explain the inference patterns of a legal LLM. The legal LLM correctly attributes the judgment of "*Assault*" to interactions involving the relevant phrases "*chased*," "*with an axe*," "*bit*," and "*slightly injured*." However, the legal LLM also uses the irrelevant phrases ("*morning*" and "*had an argument*"), and the forbidden phrases ("*hit*," "*with a shovel*," "*injuring*," and "*death*") to compute the output score of the judgment of "*Assault*." These irrelevant phrases do not directly contribute to the legal judgment result for Andy, and the forbidden phrases are actions and consequences that are not directly related to Andy, *e.g.,* actions are not taken by Andy. Obviously, the judgment should not rely on such inference patterns.

In this way, we define *reliable* and *unreliable* interaction effects for AND-OR interactions, as follows.

*For AND interactions.* Because the AND interaction $I_S^{\text{AND}}$ is activated only when all input phrases (tokens or phrases) in $S$ are present in the input legal case, the reliable interaction effect for AND interaction $R_S^{\text{AND}}$ *w.r.t.* $S$ must include relevant phrases in $\mathcal{R}$, and completely exclude forbidden phrases in $\mathcal{F}$, *i.e.,* $S \cap \mathcal{R} \neq \emptyset, S \cap \mathcal{F} = \emptyset$. Otherwise, if $S$ contains any forbidden phrases in $\mathcal{F}$, or if $S$ does not contains any relevant phrases in $\mathcal{R}$, then the AND interaction $I_S^{\text{AND}}$ represents an incorrect logic for judgment. In this way, the reliable AND interaction effects $R_S^{\text{AND}}$ and unreliable AND interaction effects $U_S^{\text{AND}}$ *w.r.t.* $S$ can be computed as follows.

$$
\begin{aligned}
\text{if} \quad S \cap \mathcal{F} = \emptyset, S \cap \mathcal{R} \neq \emptyset \quad &\text{then} \quad R_S^{\text{AND}} = I_S^{\text{AND}}, \quad U_S^{\text{AND}} = 0 \\
&\text{otherwise,} \quad R_S^{\text{AND}} = 0, \quad U_S^{\text{AND}} = I_S^{\text{AND}}
\end{aligned}
\tag{4}
$$

*For OR interactions.* The OR interaction $I_S^{\text{OR}}$ affects the LLM's output when any input variable (token or phrase) in $S$ appears in the input legal case. Therefore, we can define the reliable effect $R_S^{\text{OR}}$ as the numerical component in $I_S^{\text{OR}}$ allocated to relevant input phrases in $S \cap \mathcal{R}$. To this end, just like in (10), we uniformly allocate the OR interaction effects to all input phrases in $S$. The reliable interaction effects $R_S^{\text{OR}}$ and unreliable interactions effects $U_S^{\text{OR}}$ are those allocated to relevant variables, and those allocated to irrelevant and forbidden variables, respectively.

$$
\forall S \subseteq N, S \neq \emptyset, \quad R_S^{\text{OR}} = \frac{|S \cap \mathcal{R}|}{|S|} \cdot I_S^{\text{OR}}, \quad U_S^{\text{OR}} = \left(1 - \frac{|S \cap \mathcal{R}|}{|S|}\right) \cdot I_S^{\text{OR}}
\tag{5}
$$

In fact, such a uniform allocation of interaction effects to input phrases has sufficient theoretical supports and has been widely used, *e.g.,* being used in the computation of the Shapley value (45; 32).

In this way, according to Equation (2) with the setting $\mathbf{x}_{\text{mask}} = \mathbf{x}$, the output score of the legal judgment result $v(\mathbf{x})$ can be formulated as the sum of all reliable effects ($R_S^{\text{AND}}$ and $R_S^{\text{OR}}$) that align

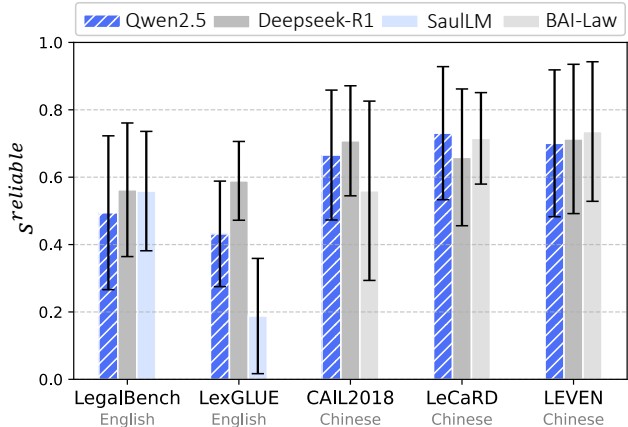

Figure 2: Ratio of reliable interaction effects (measured by $s^{\text{reliable}}$) among all the interaction patterns used by the LLM for judgment.

with human domain knowledge, and all unreliable effects ($U_S^{\text{AND}}$ and $U_S^{\text{OR}}$) that do not match human domain knowledge.

$$v(\mathbf{x}) = v(\mathbf{x}_\emptyset) + \underbrace{\sum_{S \in \Omega^{\text{AND}}} R_S^{\text{AND}} + \sum_{S \in \Omega^{\text{OR}}} R_S^{\text{OR}}}_{\text{reliable interaction effects}} + \underbrace{\sum_{S \in \Omega^{\text{AND}}} U_S^{\text{AND}} + \sum_{S \in \Omega^{\text{OR}}} U_S^{\text{OR}}}_{\text{unreliable interaction effects}} \qquad (6)$$

**Ratio of reliable interaction effects.** We design a metric to evaluate the alignment quality between the interaction patterns used by the legal LLM and human domain knowledge. Definition 1 introduces the ratio of reliable interaction effects that align with human domain knowledge.

**Definition 1** (Ratio of reliable interaction effects). *Given an LLM, the ratio of reliable interaction effects to all salient interaction effects $s^{\text{reliable}}$ is computed as follows.*

$$s^{\text{reliable}} = \frac{\sum_{S \in \Omega^{\text{AND}}} |R_S^{\text{AND}}| + \sum_{S \in \Omega^{\text{OR}}} |R_S^{\text{OR}}|}{\sum_{S \in \Omega^{\text{AND}}} |I_S^{\text{AND}}| + \sum_{S \in \Omega^{\text{OR}}} |I_S^{\text{OR}}|} \qquad (7)$$

A larger value of $s^{\text{reliable}} \in [0, 1]$ indicates that a higher proportion of interaction effects align with human domain knowledge, which means the judgment rationale of an LLM is more aligned with that of human experts.

## 3 EXPERIMENTS

In this section, we conducted experiments to evaluate the correctness of interaction patterns (inference patterns) used by the LLMs for legal judgments. Specifically, we identified and quantified the reliable interaction effects and unreliable interaction effects used by the LLM.

We evaluated the correctness of interaction patterns used by four LLMs: two general-purpose LLMs, including Qwen2.5-14B-Base (57), and Deepseek-R1-Distill-Qwen-14B (29), and two law-specific LLMs, including SaulLM-7B-Instruct (7), and BAI-Law-13B (21). Among them, SaulLM-7B-Instruct was trained on English legal corpora, while BAI-Law-13B was fine-tuned on Chinese legal corpora.

**Examining the faithfulness of the interaction-based explanation.** We conducted experiments to evaluate the sparsity property and the universal matching property of the extracted interactions in Section J. The successful verification of the two properties indicated that the intricate inference logic used by the LLM for judgment on exponentially many masked input legal cases could be faithfully mimicked by the few extracted AND-OR interactions.

### 3.1 EVALUATING THE RELIABILITY OF INTERACTIONS USED FOR JUDGMENT

The disentanglement of reliable interaction effects and unreliable interaction effects provides new perspectives to analyze the representation quality of an LLM.

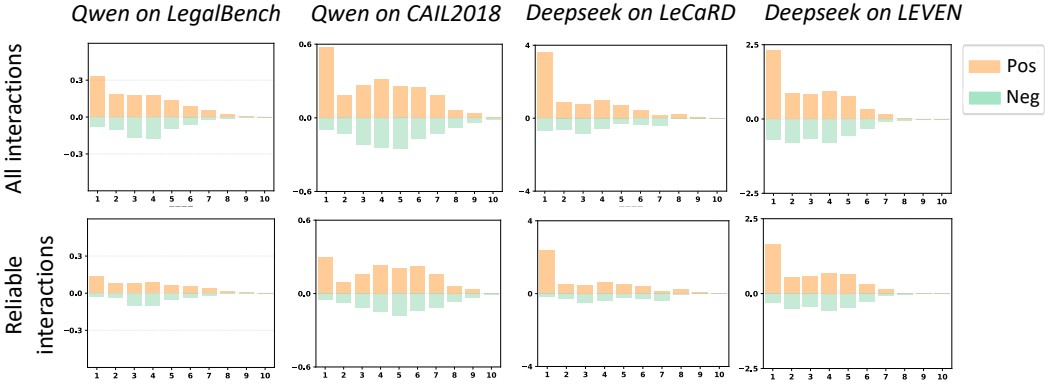

Figure 3: Distribution of all interactions over different orders (complexities) (denoted by $A^{(o),\text{pos}}$ and $A^{(o),\text{neg}}$) and that of all reliable interactions (denoted by $A^{(o),\text{pos}}_{\text{reliable}}$ and $A^{(o),\text{neg}}_{\text{reliable}}$).

**Ratio of reliable interaction effects.** Figure 2 compares the ratio of reliable interaction effects $s^{\text{reliable}}$ used by different LLMs for judgment. Specifically, for English legal tasks, we evaluated Qwen, Deepseek, and SaulLM on legal cases from the ECtHR dataset in the LexGLUE benchmark (5) and the Learned Hand Crime dataset in the LegalBench benchmark (17). For Chinese legal tasks, we evaluated Qwen, Deepseek, and BAI-Law on cases from the CAIL2018 dataset (56), the LeCaRD dataset (33), and the LEVEN dataset (58). For each task, we evaluated 100 randomly selected samples, with 10 informative input phrases chosen by two senior legal experts with over 10 years of professional experience. Then, we invited a group of 16 legal experts and volunteers to annotate each phrase as relevant, irrelevant, and forbidden phrases in the input legal case. Please refer to Section L.1 for more implemental details.

Empirical results demonstrated that neither general-purpose LLMs nor legal-domain-specific LLMs exhibited sufficient reliable interaction effects. In particular, over half of interactions represented unreasonable or even incorrect justifications for the judgment predictions. This reminded us that although current LLMs conducted correct predictions on legal tasks, their decisions relied on a large number of problematic rationales, which could introduce significant potential unfairness and risks.

**Complexity of interactions.** We analyzed the complexity of interactions used for judgment. We used the order of an interaction, *i.e.*, the number of input phrases in $|S|$, to represent the complexity of interactions. Specifically, let $A^{(o),\text{pos}} = \sum_{\text{op}\in\{\text{AND,OR}\}} \sum_{S\in\Omega^{\text{op}},|S|=o} \max(0, I^{\text{op}}_S)$ and $A^{(o),\text{neg}} = \sum_{\text{op}\in\{\text{AND,OR}\}} \sum_{S\in\Omega^{\text{op}},|S|=o} \min(0, I^{\text{op}}_S)$ to represent the strength of all positive $o$-order interactions and the strength of all negative negative $o$-order interactions. Figure 3 shows the histogram of $A^{(o),\text{pos}}$ and $A^{(o),\text{neg}}$ to represent the distribution of interactions over different orders (complexities). Similarly, we computed the distribution of reliable interactions over different orders by quantifying $A^{(o),\text{pos}}_{\text{reliable}}$ and $A^{(o),\text{neg}}_{\text{reliable}}$ on reliable interactions $\{R^{\text{AND}}_S, R^{\text{OR}}_S\}_S$ in the same manner (please see Figure 3).

As evidenced in Figure 3, the LLM consistently demonstrated a strong preference for using low-order interactions for legal judgments, regardless of whether we examined the distribution of all interactions or the distribution of only reliable interactions. The low-order interactions mainly used local patterns on few input phrases to facilitate heuristic-based inference, rather than conducting comprehensive analysis of all case factors. *This finding had challenged the prevailing assumption that LLMs possessed long-chain reasoning capabilities.*

**Significance of conflicted interaction patterns.** Besides, we also quantified the significance of conflicts between different interaction effects. We can consider positive interaction effects as supporting evidence for generating the target tokens, while negative interaction effects serve as anti-evidence. Therefore, we quantified the significance of such cancellation for interactions as $s^{\text{conflict}} = 1 - \sum_{\text{op}\in\{\text{AND,OR}\}} |\sum_{S\in\Omega^{\text{op}}} I^{op}_S| / \sum_{\text{op}\in\{\text{AND,OR}\}} \sum_{S\in\Omega^{\text{op}}} |I^{op}_S| \in [0, 1]$. Table 4 in Section L.2 shows the significance of mutual cancellation of interaction patterns. We found that roughly more than 60% effects of the interaction patterns had been mutually cancelled out. The mutually canceling interaction effects demonstrated the inherent ambiguity in an LLM's judgment. In contrast, more reliable large models typically exhibited lower cancellation level.

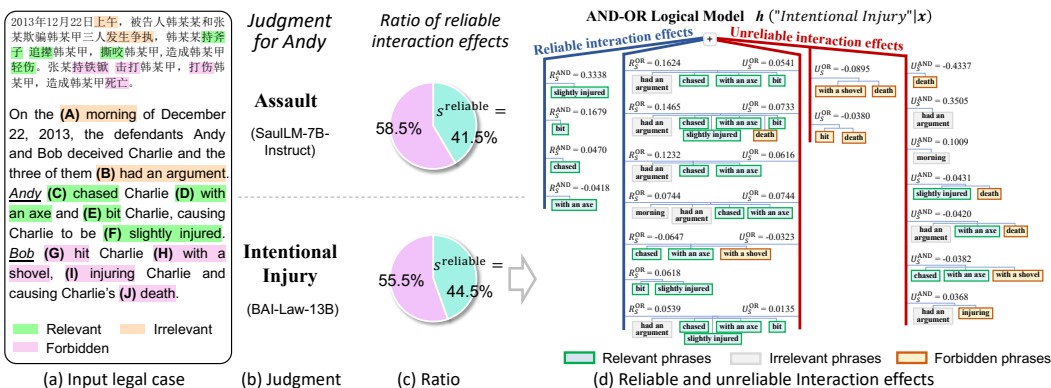

Figure 4: Visualization of judgments affected by incorrect entities' actions. (a) Irrelevant phrases were annotated in the legal case, including the time and defendant's actions that were not the direct reason for the judgment. Criminal actions of the defendant were annotated as relevant phrases. Criminal actions of the unrelated person were annotated as forbidden phrases. (b) Judgments predicted by the two legal LLMs, which were both correct according to laws of the two countries. (c,d) We quantified the reliable and unreliable interaction effects.

## 3.2  CASE STUDIES

In this subsection, we visualized the interaction patterns on specific legal cases, and identified potential representation flaws of LLMs. While not exhaustive, let us introduce three common types of potential representation flaws frequently observed in LLMs: (1) making judgments using the behavior of incorrect entities, (2) making judgments influenced by identity-based discrimination, and (3) making judgments based on semantically irrelevant phrases. Due to the limit of the page number, we analyzed legal cases of the first and second types, and put results of the third type in Section K. We tested legal LLMs SaulLM and BAI-Law make judgments on legal cases in the CAIL2018 dataset (56). For the SaulLM-7B-Instruct model, we translated the Chinese legal cases into English and performed the analyses on the translated cases to enable fair comparisons.

**Case 1: making judgments based on incorrect entities' actions.** Despite the high accuracy of legal LLMs in predicting judgment results, we observed that the legal LLMs used a significant portion of interaction patterns that were mistakenly attributed to criminal actions made by incorrect entities. In other words, the legal LLMs mistakenly used the criminal action of a person (entity) to make judgment on another unrelated person (entity). To evaluate the impact of such incorrect entity matching on both the SaulLM and BAI-Law models, we engaged legal experts to annotate misleading phrases that described incorrect defendant as the *forbidden phrases* in $\mathcal{F}$. These forbidden phrases should not influence the legal judgment for the target defendant.

Figure 4 shows the legal case, which showed Andy bit Charlie, constituting an assault, and then Bob hit Charlie with a shovel, resulting in Charlie's death. Here, when the legal LLMs judged the actions of Andy, input phrases such as "*hit*," "*with a shovel*," "*injuring*," and "*death*" were annotated as forbidden phrases in $\mathcal{F}$, because these phrases described Bob's actions and consequences and were not directly related to Andy. We observed that the SaulLM *did* use several interaction patterns which aligned with legal experts' domain knowledge for the judgment in Figure 1. For example, an AND interaction pattern $S_1 = \{\text{"}slightly\ injured\text{"}\}$, an AND interaction pattern $S_2 = \{\text{"}bit\text{"}\}$, and an OR interaction pattern $S_3 = \{\text{"}bit\text{"}, \text{"}slightly\ injured\text{"}\}$ contributed salient reliable interaction effects $R_{S_1}^{\text{AND}} = 0.47$, $R_{S_2}^{\text{AND}} = 0.33$, and $R_{S_3}^{\text{OR}} = 0.10$, respectively, to the confidence score $v(\text{"}Assault\text{"}|\mathbf{x})$ of the judgment "*Assault*" for Andy. However, the legal LLM also used a significant portion of problematic interaction patterns that based on an incorrect entity's actions. For example, three AND interaction patterns $S_4 = \{\text{"}death\text{"}\}$, $S_5 = \{\text{"}with\ a\ shovel\text{"}\}$, and $S_6 = \{\text{"}injuring\text{"}\}$ that described Bob's actions and consequences contributed unreliable interaction effects $U_{S_4}^{\text{AND}} = -1.04$, $U_{S_5}^{\text{AND}} = 0.93$ and $U_{S_6}^{\text{AND}} = 0.19$ to the confidence score of the judgment "*Assault*" for Andy, respectively. In sum, the SaulLM model only used a ratio of $s^{\text{reliable}} = 41.5\%$ reliable interaction effects for the legal judgment. This reflected a representation flaw, *i.e.*, the LLM tended to memorize the sensitive tokens, such as the weapons, alongside the legal judgment results, rather than understand the true logic in the input prompt, *e.g.*, identifying *who* performed *which* actions.

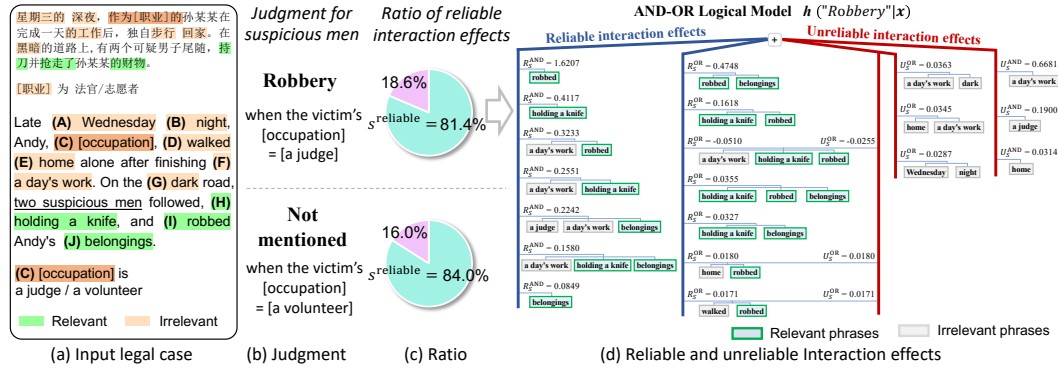

Figure 5: Visualization of judgments biased by discrimination in identity. (a) Irrelevant phrases were annotated in the legal case, including the occupation, time and actions that are not the direct reason for the judgment. Criminal actions of the defendant were annotated as relevant phrases. (b) The SaulLM-7B-Instruct model predicted the judgment based on the legal case with different occupations. (c,d) We quantified the reliable and unreliable interaction effects.

In comparison, we evaluated the above legal case on the BAI-Law model, as shown in Figure 4. The BAI-Law model used a bit higher ratio of $s^{\text{reliable}} = 44.5\%$ reliable interaction effects. Many interaction patterns used by the BAI-Law-13B model were also used by the SaulLM model, such as an AND interaction pattern $S_1 = \{\text{"slightly injured"}\}$, and an AND interaction pattern $S_2 = \{\text{"bit"}\}$, and an OR interaction pattern $S_3 = \{\text{"bit"}, \text{"slightly injured"}\}$ contributed salient reliable interaction effects $R_{S_1}^{\text{AND}} = 0.33$, $R_{S_2}^{\text{AND}} = 0.17$, and $R_{S_3}^{\text{OR}} = 0.06$ to the confidence score $v(\text{"Intentional Injury"}|\mathbf{x})$ of the judgment "*Intentional Injury*" for Andy, respectively. This indicated that these two legal LLMs *did* identify some direct reasons for the legal judgment. However, the BAI-LAW-13B model also primarily relied on unreliable interaction effects for the legal judgment on Andy, such as an AND interaction pattern $S_4 = \{\text{"death"}\}$, which included forbidden phrases for the consequence of Bob's actions, to contribute unreliable interaction effects $U_{S_4}^{\text{AND}} = -0.43$ to the confidence score of the judgment "*Intentional Injury*" for Andy. Additional examples of making judgments based on incorrect entities' actions are provided in Section L.4.

**Case 2: discrimination in identity may affect judgments.** We observed that the legal LLMs used interaction patterns that were attributed to the occupation information. This would lead to a significant occupation bias. More interestingly, we observed that when we replaced the occupation phrase with another occupation phrase, the unreliable interaction effect containing the occupation phrase would be significant changed. This indicates a common identity bias problem, because similar bias may also happen on other identities (*e.g.*, age, gender, education level, and marital status).

Figure 5 shows the legal case, in which Andy was robbed of his belongings by two suspicious men. The SaulLM used several interaction patterns that aligned with legal experts' domain knowledge for the legal judgment, *e.g.*, an AND interaction pattern $S_1 = \{\text{"robbed"}\}$, and an OR interaction pattern $S_2 = \{\text{"robbed"}, \text{"belongings"}\}$, and an OR interaction pattern $S_3 = \{\text{"holding a knife"}, \text{"robbed"}\}$ contributed salient reliable interaction effects to the confidence score $v(\text{"Robbery"}|\mathbf{x})$ of the judgment "*Robbery*." However, the legal LLM also used problematic interaction patterns, *i.e.*, an AND interaction pattern $S_4 = \{\text{"a judge"}\}$ for the occupation information contributed salient unreliable interaction effects $U_{S_4}^{\text{AND}} = 0.19$ to boost the output score of the judgment.

More interestingly, if we substituted Andy's occupation from the phrase "*a judge*" to "*a volunteer*," the interaction pattern $S_5 = \{\text{"[occupation]"}, \text{"a day's work"}, \text{"belongings"}\}$ decreased its reliable interaction effects from $R_{S_5}^{\text{AND}} = 0.22$ to $R_{S_5}^{\text{AND}} = 0.06$ (see Figure 11 in Appendix). The interaction patterns containing the occupation phrase were important factors that changed the legal judgment result from "*Robbery*" to "*Not mentioned*." We verified similar phenomena on different occupations, *e.g.*, substituting the occupation phrase with law-related occupations such as "*a lawyer*" and "*a policeman*" also maintained the judgment result, while the other occupations such as "*a programmer*" changed the judgment to "*Not mentioned*." Please see Section L.5 for reliable and unreliable interaction effects for all these occupations. This suggested considerable occupation bias. In comparison, we evaluated the same legal case on the BAI-Law in Section L.5. This experiment showed the potential of our method to identify the identity (*e.g.*, occupation) bias used by the LLM.

## 4 Conclusions and discussion

In this paper, we proposed a method to evaluate the correctness of the detailed inference patterns used by an LLM. The universal matching property and the sparsity property of interactions provide mathematical support for the faithfulness of interaction-based explanations. Thus, in this paper, we designed new metrics to identify and quantify reliable and unreliable interaction effects. Experiments showed that the legal LLMs often used a significant portion of problematic interaction patterns to make judgments, even when the legal judgment prediction appeared correct. The evaluation of the alignment between the interaction patterns of LLMs and human domain knowledge has broader implications for high-stake tasks, such as finance and healthcare data analytics, although we focus on legal LLMs as a case study.

### Acknowledgments

This work is partially supported by the National Nature Science Foundation of China (92370115, 62276165), and Shanghai Natural Science Foundation (24ZR1491700).

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

## A  THE USE OF LARGE LANGUAGE MODELS (LLMS)

In this paper, large language models (LLMs) were used solely for partial language refinement.

## B  RELATED WORK

Previous works have evaluated different aspects of trustworthiness and safety in LLMs, including factuality and hallucination problems, value alignment, and susceptibility to attacks. First, the evaluation of factuality refers to whether the language generalization results of LLMs align with the verifiable facts (28; 37; 51). Hallucination in LLMs typically arises when the generated results contradict the source material or cannot be verified from the provided input (14; 34; 19; 34; 20; 12; 23). Second, value alignment aims to ensure an LLM to behave in accordance with human intentions and values (25; 52; 22), which is another classical perspective for evaluating the bias and safety of LLMs. Recent studies have used Supervised Fine-Tuning (SFT) (38; 37) and Reinforcement Learning from Human Feedback (RLHF) (38; 37; 49) to align LLM's behavior with human expectations. Third, susceptibility to attacks is also another significant concern for LLMs. Recent studies have shown that even the latest LLMs remain vulnerable to adversarial inputs to generate harmful content (62; 54; 3; 37), which is also known as "jailbreaks."

However, above evaluation methods mainly focus on the quality or correctness of output results of LLMs. The high accuracy of the LLM usually makes the evaluation a long-tail search for incorrect results.

In comparison, our evaluation approach examines the correctness of internal interaction patterns. Even when the LLM outputs correct results on a testing sample, experimental results show that more than a half detailed interaction patterns encoded by the legal LLM may still represent chaotic features. Thus, we can consider the interaction pattern as a much more efficient evaluation strategy. Our goal is to enhance the trustworthiness of the LLMs, particularly in high-stake tasks. Essentially, two types of evaluation strategies can be roughly analogized to the distinction between procedural fairness and outcome fairness.

**Reviewing the development of the interaction explanation theory.** A representative approach in explainable AI was to explain the interactions between input variables (47; 50). Based on the game theory, (39) first used the Harsanyi dividend (18) to quantify the the interaction effect between input variables encoded by the DNN. Then, (26) discovered and (42) further proved that the output scores of DNNs can be faithfully explained as a small number of interaction patterns between input variables. Furthermore, (9; 30; 41) further demonstrated the representation bottleneck of different neural networks from the perspective of interactions, *i.e.*, proving interactions of specific complexities are difficult for specific DNNs to encode. (61) explored the relationship between the complexity of interactions and the generalization power of DNNs. Additionally, (10) proved that the interaction theory provides a unified explanation for mathematical mechanisms of 14 most widely used attribution methods, including Grad-CAM (44), Integrated Gradients (48), and Shapley values (45; 32). (60) proved that the interaction theory provides a unified explanation for the shared mathematical mechanism of 12 classical transferability-boosting methods.

## C  PROOF OF THEOREM

**Theorem 1** (Universal matching property)  When scalar weights in the logical model are set to $\forall S \subseteq N, I_S^{\text{AND}} \stackrel{\text{def}}{=} \sum_{T \subseteq S}(-1)^{|S|-|T|} v_{\text{and}}(\mathbf{x}_T)$ and $I_S^{\text{OR}} \stackrel{\text{def}}{=} -\sum_{T \subseteq S}(-1)^{|S|-|T|} v_{\text{or}}(\mathbf{x}_{N \setminus T})$, subject to the requirement $v_{\text{and}}(\mathbf{x}_T) + v_{\text{or}}(\mathbf{x}_T) = v(\mathbf{x}_T)$, then we have $\forall T \subseteq N, v(\mathbf{x}_T) = h(\mathbf{x}_T)$.

In other words, we have to prove the following theorem.

Given an input sample $\mathbf{x}$, the network output score $v(\mathbf{x}_T) \in \mathbb{R}$ on each masked sample $\{\mathbf{x}_T | T \subseteq N\}$ can be well matched by a surrogate logical model $h(\mathbf{x}_T)$ on each masked sample $\{\mathbf{x}_T | T \subseteq N\}$. The surrogate logical model $h(\mathbf{x}_T)$ uses the sum of AND interactions and OR interactions to accurately

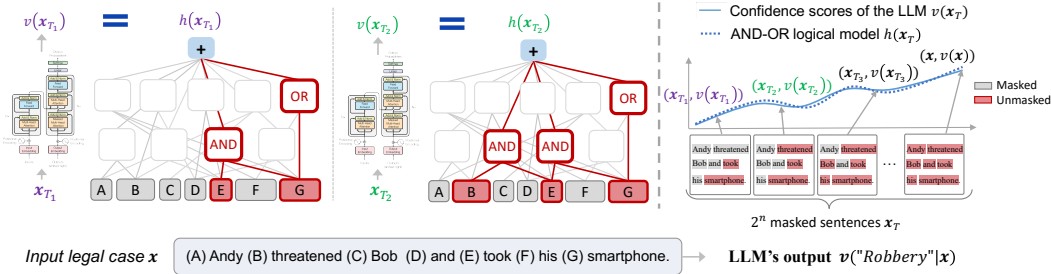

Figure 6: Theorem 1 proves that the AND-OR logical model $h(\cdot)$ can accurately match the confidence score of the LLM's outputs $v(\cdot)$ when we augment the input prompt $\mathbf{x}$ by enumerating its all $2^n$ masked states. Here, the left figure shows that the masked input prompt $\mathbf{x}_{T_1}$ with two unmasked token "*took*" and "*smartphone*" activates an AND interaction pattern $S = \{\text{"took", "smartphone"}\}$ and an OR interaction pattern $S = \{\text{"his", "smartphone"}\}$, and they contribute numerical values (interaction effects) to the logical model $h(\mathbf{x}_{T_1})$. The right figure shows that the logical model can always match the LLM's outputs on all masked states of the input prompt, $\forall T \subseteq N, h(\text{"Robbery"}|\mathbf{x}_T) = v(\text{"Robbery"}|\mathbf{x}_T)$.

fit the network output score $v(\mathbf{x}_T)$.

$$\forall T \subseteq N, v(\mathbf{x}_T) = h(\mathbf{x}_T).$$

$$h(\mathbf{x}_T) = v(\mathbf{x}_\emptyset) + \sum_{S \subseteq N, S \neq \emptyset} \mathbb{1}_{\text{AND}}(S|\mathbf{x}_T) \cdot I_S^{\text{AND}} + \sum_{S \subseteq N, S \neq \emptyset} \mathbb{1}_{\text{OR}}(S|\mathbf{x}_T) \cdot I_S^{\text{OR}}$$

$$= \underbrace{v(\mathbf{x}_\emptyset) + \sum_{S \subseteq T, S \neq \emptyset} I_S^{\text{AND}}}_{v_{\text{and}}(\mathbf{x}_T)} + \underbrace{\sum_{S \subseteq N, S \cap T \neq \emptyset} I_S^{\text{OR}}}_{v_{\text{or}}(\mathbf{x}_T)}$$

(8)

*Proof.* Let us set a surrogate logical model $h(\mathbf{x}_T) = v(\mathbf{x}_T), \forall T \subseteq N$, which utilizes the sum of AND interactions $I_S^{\text{AND}} = \sum_{T \subseteq S} (-1)^{|S|-|T|} v_{\text{and}}(\mathbf{x}_T)$ and OR interactions $I_S^{\text{OR}} = -\sum_{T \subseteq S} (-1)^{|S|-|T|} v_{\text{or}}(\mathbf{x}_{N \setminus T})$ to fit the network output score $v(\mathbf{x}_T)$, *i.e.*, $v_{\text{and}}(\mathbf{x}_T) + v_{\text{or}}(\mathbf{x}_T) = v(\mathbf{x}_T)$.

To be specific, (1) we use the sum of AND interactions $I_S^{\text{AND}}$ to compute the component for AND interactions $v_{\text{and}}(\mathbf{x}_T)$, *i.e.*, $v_{\text{and}}(\mathbf{x}_T) = \sum_{S \subseteq T} I_S^{\text{AND}}$. (2) Then, we use the sum of OR interactions $I_S^{\text{OR}}$ to compute the component for OR interactions $v_{\text{or}}(\mathbf{x}_T)$, *i.e.*, $v_{\text{or}}(\mathbf{x}_T) = \sum_{S \subseteq N, S \cap T \neq \emptyset} I_S^{\text{OR}}$. (3) Finally, we use the surrogate logical model $h(\cdot)$ (which uses the sum of AND interactions and OR interactions) to fit the network output score $v(\cdot)$, *i.e.*, $\forall T \subseteq N, v_{\text{and}}(\mathbf{x}_T) + v_{\text{or}}(\mathbf{x}_T) = v(\mathbf{x}_T) = h(\mathbf{x}_T)$.

### (1) Universal matching property of AND interactions.

(39) first used the Harsanyi dividend $I_S^{\text{AND}}$ in the cooperative game theory (18) to state the universal matching property of AND interactions. The output score of a well-trained DNN on all $2^n$ masked samples $\{\mathbf{x}_T | T \subseteq N\}$ could be universally explained by the all interaction patterns in $T \subseteq N$, *i.e.*, $\forall T \subseteq N, v_{\text{and}}(\mathbf{x}_T) = \sum_{S \subseteq T} I_S^{\text{AND}}$.

Specifically, the AND interaction (as known as Harsanyi dividend) is defined as $I_S^{\text{AND}} := \sum_{L \subseteq S} (-1)^{|S|-|L|} v_{\text{and}}(\mathbf{x}_L)$. To compute the sum of AND interactions $\forall T \subseteq N, \sum_{S \subseteq T} I_S^{\text{AND}} = \sum_{S \subseteq T} \sum_{L \subseteq S} (-1)^{|S|-|L|} v_{\text{and}}(\mathbf{x}_L)$, we first exchange the order of summation of the set $L \subseteq S \subseteq T$ and the set $S \supseteq L$. That is, we compute all linear combinations of all sets $S$ containing $L$ with respect to the model outputs $v_{\text{and}}(\mathbf{x}_L)$, given a set of input phrases $L$, *i.e.*, $\sum_{S: L \subseteq S \subseteq T} (-1)^{|S|-|L|} v_{\text{and}}(\mathbf{x}_L)$. Then, we compute all summations over the set $L \subseteq T$.

In this way, we can compute them separately for different cases of $L \subseteq S \subseteq T$. In the following, we consider the cases (1) $L = S = T$, and (2) $L \subseteq S \subseteq T, L \neq T$, respectively.

(1) When $L = S = T$, the linear combination of all subsets $S$ containing $L$ with respect to the model output $v_{\text{and}}(\mathbf{x}_L)$ is $(-1)^{|T|-|T|} v_{\text{and}}(\mathbf{x}_L) = v_{\text{and}}(\mathbf{x}_L)$.

(2) When $L \subseteq S \subseteq T, L \neq T$, the linear combination of all subsets $S$ containing $L$ with respect to the model output $v_{\text{and}}(\mathbf{x}_L)$ is $\sum_{S:L\subseteq S\subseteq T}(-1)^{|S|-|L|}v_{\text{and}}(\mathbf{x}_L)$. For all sets $S : T \supseteq S \supseteq L$, let us consider the linear combinations of all sets $S$ with number $|S|$ for the model output $v_{\text{and}}(\mathbf{x}_L)$, respectively. Let $m := |S|-|L|, (0 \leq m \leq |T|-|L|)$, then there are a total of $C_{|T|-|L|}^m$ combinations of all sets $S$ of order $|S|$. Thus, given $L$, accumulating the model outputs $v_{\text{and}}(\mathbf{x}_L)$ corresponding to all $S \supseteq L$, then $\sum_{S:L\subseteq S\subseteq T}(-1)^{|S|-|L|}v_{\text{and}}(\mathbf{x}_L) = v_{\text{and}}(\mathbf{x}_L) \cdot \underbrace{\sum_{m=0}^{|T|-|L|} C_{|T|-|L|}^m(-1)^m}_{=0} = 0$.

Please see the complete derivation of the following formula.

$$
\begin{aligned}
\sum_{S\subseteq T} I_S^{\text{AND}} &= \sum_{S\subseteq T}\sum_{L\subseteq S}(-1)^{|S|-|L|}v_{\text{and}}(\mathbf{x}_L) \\
&= \sum_{L\subseteq T}\sum_{S:L\subseteq S\subseteq T}(-1)^{|S|-|L|}v_{\text{and}}(\mathbf{x}_L) \\
&= \underbrace{v_{\text{and}}(\mathbf{x}_T)}_{L=T} + \sum_{L\subseteq T,L\neq T} v_{\text{and}}(\mathbf{x}_L) \cdot \underbrace{\sum_{m=0}^{|T|-|L|} C_{|T|-|L|}^m(-1)^m}_{=0} \\
&= v_{\text{and}}(\mathbf{x}_T).
\end{aligned}
\tag{9}
$$

Furthermore, we can understand the above equation in a physical sense. Given a masked sample $\mathbf{x}_T$, if $\mathbf{x}_T$ triggers an AND relationship $S$ (the co-appearance of all input phrases in $S$), then $S \subseteq T$. Thus, we accumulate the interaction effects $I_S^{\text{AND}}$ of any AND relationship $S$ triggered by $\mathbf{x}_T$ as follows,

$$
\begin{aligned}
v(\mathbf{x}_\emptyset) &+ \sum_{S\subseteq N,S\neq\emptyset} \mathbb{1}_{\text{AND}}(S|\mathbf{x}_T) \cdot I_S^{\text{AND}} \\
&= v(\mathbf{x}_\emptyset) + \sum_{S\subseteq T,S\neq\emptyset} I_S^{\text{AND}} \\
&= \sum_{S\subseteq T} I_S^{\text{AND}} \\
&= v_{\text{and}}(\mathbf{x}_T).
\end{aligned}
\tag{10}
$$

**(2) Universal matching property of OR interactions.**

According to the definition of OR interactions, we will derive that $\forall T \subseteq N, v_{\text{or}}(\mathbf{x}_T) = \sum_{S\subseteq N,S\cap T\neq\emptyset} I_S^{\text{OR}}, s.t., I_\emptyset^{\text{OR}} = v_{\text{or}}(\mathbf{x}_\emptyset) = 0$.

Specifically, the OR interaction is defined as $I_S^{\text{OR}} := -\sum_{L\subseteq S}(-1)^{|S|-|L|}v_{\text{or}}(\mathbf{x}_{N\setminus L})$. To compute the sum of OR interactions $\forall T \subseteq N, \sum_{S\subseteq N,S\cap T\neq\emptyset} I_S^{\text{OR}} = \sum_{S\subseteq N,S\cap T\neq\emptyset}\left[-\sum_{L\subseteq S}(-1)^{|S|-|L|}v_{\text{or}}(\mathbf{x}_{N\setminus L})\right]$, we first exchange the order of summation of the set $L \subseteq S \subseteq N$ and the set $S \cap T \neq \emptyset$. That is, we compute all linear combinations of all sets $S$ containing $L$ with respect to the model outputs $v_{\text{or}}(\mathbf{x}_{N\setminus L})$, given a set of input phrases $L$, i.e., $\sum_{S\cap T\neq\emptyset,N\supseteq S\supseteq L}(-1)^{|S|-|L|}v_{\text{or}}(\mathbf{x}_{N\setminus L})$. Then, we compute all summations over the set $L \subseteq N$.

In this way, we can compute them separately for different cases of $L \subseteq S \subseteq N, S \cap T \neq \emptyset$. In the following, we consider the cases (1) $L = N \setminus T$, (2) $L = N$, (3) $L \cap T \neq \emptyset, L \neq N$, and (4) $L \cap T = \emptyset, L \neq N \setminus T$, respectively.

(1) When $L = N \setminus T$, the linear combination of all subsets $S$ containing $L$ with respect to the model output $v_{\text{or}}(\mathbf{x}_{N\setminus L})$ is $\sum_{S\cap T\neq\emptyset,S\supseteq L}(-1)^{|S|-|L|}v_{\text{or}}(\mathbf{x}_{N\setminus L}) = \sum_{S\cap T\neq\emptyset,S\supseteq L}(-1)^{|S|-|L|}v_{\text{or}}(\mathbf{x}_T)$. For all sets $S \supseteq L, S\cap T \neq \emptyset$ (then $S \neq N\setminus T, S \neq L$), let us consider the linear combinations of all sets $S$ with number $|S|$ for the model output $v_{\text{or}}(\mathbf{x}_T)$, respectively. Let $|S'| := |S|-|L|, (1 \leq |S'| \leq |T|)$, then there are a total of $C_{|T|}^{|S'|}$ combinations of all sets $S$ of order $|S|$. Thus, given $L$, accumulating the model outputs $v_{\text{or}}(\mathbf{x}_T)$ corresponding to all $S \supseteq L$, then $\sum_{S\cap T\neq\emptyset,S\supseteq L}(-1)^{|S|-|L|}v_{\text{or}}(\mathbf{x}_{N\setminus L}) = v_{\text{or}}(\mathbf{x}_T) \cdot \underbrace{\sum_{|S'|=1}^{|T|} C_{|T|}^{|S'|}(-1)^{|S'|}}_{=-1} = -v_{\text{or}}(\mathbf{x}_T)$.

(2) When $L = N$ (then $S = N$), the linear combination of all subsets $S$ containing $L$ with respect to the model output $v_{\text{or}}(\mathbf{x}_{N\setminus L})$ is $\sum_{S\cap T\neq\emptyset, S\supseteq L}(-1)^{|S|-|L|}v_{\text{or}}(\mathbf{x}_{N\setminus L}) = (-1)^{|N|-|N|}v_{\text{or}}(\mathbf{x}_\emptyset) = v_{\text{or}}(\mathbf{x}_\emptyset) = 0$, $(I_\emptyset^{\text{OR}} = v_{\text{or}}(\mathbf{x}_\emptyset) = 0)$.

(3) When $L \cap T \neq \emptyset, L \neq N$, the linear combination of all subsets $S$ containing $L$ with respect to the model output $v_{\text{or}}(\mathbf{x}_{N\setminus L})$ is $\sum_{S\cap T\neq\emptyset,S\supseteq L}(-1)^{|S|-|L|}v_{\text{or}}(\mathbf{x}_{N\setminus L})$. For all sets $S \supseteq L, S\cap T \neq \emptyset$, let us consider the linear combinations of all sets $S$ with number $|S|$ for the model output $v_{\text{or}}(\mathbf{x}_T)$, respectively. Let us split $|S| - |L|$ into $|S'|$ and $|S''|$, $i.e.,|S| - |L| = |S'| + |S''|$, where $S' = \{i | i \in S, i \notin L, i \in N \setminus T\}$, $S'' = \{i | i \in S, i \notin L, i \in T\}$ (then $0 \leq |S''| \leq |T| - |T \cap L|$) and $S' + S'' + L = S$. In this way, there are a total of $C_{|T|-|T\cap L|}^{|S''|}$ combinations of all sets $S''$ of order $|S''|$. Thus, given $L$, accumulating the model outputs $v_{\text{or}}(\mathbf{x}_{N\setminus L})$ corresponding to all $S \supseteq L$, then $\sum_{S\cap T\neq\emptyset,S\supseteq L}(-1)^{|S|-|L|}v_{\text{or}}(\mathbf{x}_{N\setminus L}) = v_{\text{or}}(\mathbf{x}_{N\setminus L}) \cdot \sum_{S'\subseteq N\setminus T\setminus L} \underbrace{\sum_{|S''|=0}^{|T|-|T\cap L|} C_{|T|-|T\cap L|}^{|S''|}(-1)^{|S'|+|S''|}}_{=0} = 0$.

(4) When $L \cap T = \emptyset, L \neq N \setminus T$, the linear combination of all subsets $S$ containing $L$ with respect to the model output $v_{\text{or}}(\mathbf{x}_{N\setminus L})$ is $\sum_{S:S\cap T\neq\emptyset,S\supseteq L}(-1)^{|S|-|L|}v_{\text{or}}(\mathbf{x}_{N\setminus L})$. Similarly, let us split $|S|-|L|$ into $|S'|$ and $|S''|$, $i.e.,|S| - |L| = |S'| + |S''|$, where $S' = \{i | i \in S, i \notin L, i \in N \setminus T\}$, $S'' = \{i | i \in S, i \in T\}$ (then $0 \leq |S''| \leq |T|$) and $S' + S'' + L = S$. In this way, there are a total of $C_{|T|}^{|S''|}$ combinations of all sets $S''$ of order $|S''|$. Thus, given $L$, accumulating the model outputs $v_{\text{or}}(\mathbf{x}_{N\setminus L})$ corresponding to all $S \supseteq L$, then $\sum_{S\cap T\neq\emptyset,S\supseteq L}(-1)^{|S|-|L|}v_{\text{or}}(\mathbf{x}_{N\setminus L}) = v_{\text{or}}(\mathbf{x}_{N\setminus L}) \cdot \sum_{S'\subseteq N\setminus T\setminus L} \underbrace{\sum_{|S''|=0}^{|T|} C_{|T|}^{|S''|}(-1)^{|S'|+|S''|}}_{=0} = 0$.

Please see the complete derivation of the following formula.

$$
\begin{aligned}
\sum_{S\subseteq N,S\cap T\neq\emptyset} I_S^{\text{OR}} &= \sum_{S\subseteq N,S\cap T\neq\emptyset}\left[-\sum_{L\subseteq S}(-1)^{|S|-|L|}v_{\text{or}}(\mathbf{x}_{N\setminus L})\right] \\
&= -\sum_{L\subseteq N}\sum_{S\cap T\neq\emptyset,N\supseteq S\supseteq L}(-1)^{|S|-|L|}v_{\text{or}}(\mathbf{x}_{N\setminus L}) \\
&= -\left[\sum_{|S'|=1}^{|T|}C_{|T|}^{|S'|}(-1)^{|S'|}\right]\cdot\underbrace{v_{\text{or}}(\mathbf{x}_T)}_{L=N\setminus T}-\underbrace{v_{\text{or}}(\mathbf{x}_\emptyset)}_{L=N} \\
&\quad -\sum_{L\cap T\neq\emptyset,L\neq N}\left[\sum_{S'\subseteq N\setminus T\setminus L}\left(\sum_{|S''|=0}^{|T|-|T\cap L|}C_{|T|-|T\cap L|}^{|S''|}(-1)^{|S'|+|S''|}\right)\right]\cdot v_{\text{or}}(\mathbf{x}_{N\setminus L}) \\
&\quad -\sum_{L\cap T=\emptyset,L\neq N\setminus T}\left[\sum_{S'\subseteq N\setminus T\setminus L}\left(\sum_{|S''|=0}^{|T|}C_{|T|}^{|S''|}(-1)^{|S'|+|S''|}\right)\right]\cdot v_{\text{or}}(\mathbf{x}_{N\setminus L}) \\
&= -(-1)\cdot v_{\text{or}}(\mathbf{x}_T)-v_{\text{or}}(\mathbf{x}_\emptyset)-\sum_{L\cap T\neq\emptyset,L\neq N}\left[\sum_{S'\subseteq N\setminus T\setminus L}0\right]\cdot v_{\text{or}}(\mathbf{x}_{N\setminus L}) \\
&\quad -\sum_{L\cap T=\emptyset,L\neq N\setminus T}\left[\sum_{S'\subseteq N\setminus T\setminus L}0\right]\cdot v_{\text{or}}(\mathbf{x}_{N\setminus L}) \\
&= v_{\text{or}}(\mathbf{x}_T)
\end{aligned}
\tag{11}
$$

Furthermore, we can understand the above equation in a physical sense. Given a masked sample $\mathbf{x}_T$, if $\mathbf{x}_T$ triggers an OR relationship $S$ (the presence of any input variable in $S$), then $S\cap T \neq \emptyset, S \subseteq N$.

Thus, we accumulate the interaction effects $I_S^{\text{OR}}$ of any OR relationship $S$ triggered by $\mathbf{x}_T$ as follows,

$$
\begin{aligned}
&\sum_{S \subseteq N, S \neq \emptyset} \mathbb{1}_{\text{OR}}(S|\mathbf{x}_T) \cdot I_S^{\text{OR}} \\
&= \sum_{S \subseteq N, S \cap T \neq \emptyset} I_S^{\text{OR}} \\
&= v_{\text{or}}(\mathbf{x}_T).
\end{aligned}
\tag{12}
$$

**(3) Universal matching property of AND-OR interactions.**

With the universal matching property of AND interactions and the universal matching property of OR interactions, we can easily get $v(\mathbf{x}_T) = h(\mathbf{x}_T) = v_{\text{and}}(\mathbf{x}_T) + v_{\text{or}}(\mathbf{x}_T) = v(\mathbf{x}_\emptyset) + \sum_{S \subseteq T, S \neq \emptyset} I_S^{\text{AND}} + \sum_{S \subseteq N, S \cap T \neq \emptyset} I_S^{\text{OR}}$, thus, we obtain the universal matching property of AND-OR interactions.

$\square$

## D   SPARSITY PROPERTY OF INTERACTIONS

The surrogate logical model $h(\mathbf{x}_T)$ on each randomly masked sample $\mathbf{x}_T, T \subseteq N$ mainly uses the sum of a small number of salient AND interactions in $\Omega^{\text{AND}}$ and salient OR interactions in $\Omega^{\text{OR}}$ to approximate the network output score $v(\mathbf{x}_T)$.

$$
v(\mathbf{x}_T) = h(\mathbf{x}_T) \approx v(\mathbf{x}_\emptyset) + \sum_{S \in \Omega^{\text{AND}}} \mathbb{1}_{\text{AND}}(S|\mathbf{x}_T) \cdot I_S^{\text{AND}} + \sum_{S \in \Omega^{\text{OR}}} \mathbb{1}_{\text{OR}}(S|\mathbf{x}_T) \cdot I_S^{\text{OR}}
\tag{13}
$$

*Proof.* (42) have proven that under some common conditions[6], the confidence score $v_{\text{and}}(\mathbf{x}_T)$ of a well-trained DNN on all $2^n$ masked samples $\{\mathbf{x}_T|T \subseteq N\}$ could be universally approximated by a small number of AND interactions $T \in \Omega^{\text{AND}}$ with salient interaction effects $I_S^{\text{AND}}$, *s.t.*, $|\Omega^{\text{AND}}| \ll 2^n$, *i.e.*, $\forall T \subseteq N, v_{\text{and}}(\mathbf{x}_T) = \sum_{S \subseteq T} I_S^{\text{AND}} \approx \sum_{S \subseteq T: S \in \Omega^{\text{AND}}} I_S^{\text{AND}}$.

According to Equation (10), $v_{\text{and}}(\mathbf{x}_T) = \sum_{S \subseteq T} I_S^{\text{AND}} = v(\mathbf{x}_\emptyset) + \sum_{S \subseteq N, S \neq \emptyset} \mathbb{1}_{\text{AND}}(S|\mathbf{x}_T) \cdot I_S^{\text{AND}}$. Therefore, $v_{\text{and}}(\mathbf{x}_T) \approx v(\mathbf{x}_\emptyset) + \sum_{S \in \Omega^{\text{AND}}} \mathbb{1}_{\text{AND}}(S|\mathbf{x}_T) \cdot I_S^{\text{AND}}$.

Besides, as proven in Section H, the OR interaction can be considered as a specific AND interaction. Thus, the confidence score $v_{\text{or}}(\mathbf{x}_T)$ of a well-trained DNN on all $2^n$ masked samples $\{\mathbf{x}_T|T \subseteq N\}$ could be universally approximated by a small number of OR interactions $T \in \Omega^{\text{OR}}$ with salient interaction effects $I_S^{\text{OR}}$, *s.t.*, $|\Omega^{\text{OR}}| \ll 2^n$. Similarly, $v_{\text{or}}(\mathbf{x}_T) = \sum_{S \subseteq N, S \neq \emptyset} \mathbb{1}_{\text{OR}}(S|\mathbf{x}_T) \cdot I_S^{\text{OR}} \approx \sum_{S \in \Omega^{\text{OR}}} \mathbb{1}_{\text{OR}}(S|\mathbf{x}_T) \cdot I_S^{\text{OR}}$.

In this way, the surrogate logical model $h(\mathbf{x}_T)$ on each randomly masked sample $\mathbf{x}_T, T \subseteq N$ mainly uses the sum of a small number of salient AND interactions and salient OR interactions to approximate the network output score $v(\mathbf{x}_T)$, *i.e.*, $v(\mathbf{x}_T) = h(\mathbf{x}_T) = v_{\text{and}}(\mathbf{x}_T) + v_{\text{or}}(\mathbf{x}_T) \approx v(\mathbf{x}_\emptyset) + \sum_{S \in \Omega^{\text{AND}}} \mathbb{1}_{\text{AND}}(S|\mathbf{x}_T) \cdot I_S^{\text{AND}} + \sum_{S \in \Omega^{\text{OR}}} \mathbb{1}_{\text{OR}}(S|\mathbf{x}_T) \cdot I_S^{\text{OR}}$.

$\square$

## E   THE CORRECTNESS OF THE DETAILED INFERENCE PATTERNS OF AN LLM

Unlike traditional studies focused on the correctness of language generation results, this paper is driven by a different motivation, *i.e.*, evaluating the correctness of the detailed inference patterns of an

---

[6]There are three assumptions. (1) The high order derivatives of the DNN output with respect to the input phrases are all zero. (2) The DNN works well on the masked samples, and yield higher confidence when the input sample is less masked. (3) The confidence of the DNN does not drop significantly on the masked samples.

LLM behind its seemingly correct outputs. Although previous studies have been proposed to evaluate the performance of LLMs, rigorously evaluating the reliability of their inference patterns requires theoretically grounded mechanistic explanations, which is an area that remains unexplored. Thanks to advances in Explainable AI, we can use a set of interactions between input features to faithfully represent the inference score of a deep network. However, despite these theoretical achievements, it remains unknown (1) how many problematic interactions are modeled in LLMs (e.g., legal LLMs), and (2) to what extent these interactions influence legal judgments.

In this paper, we quantified these interactions and conducted experiments across different LLMs and datasets. We found that,

• Over half of the interactions modeled by LLMs actually represent clearly unreasonable or even incorrect justifications for their predictions.

• LLMs tend to use simple interactions of local tokens to guess judgments.

• LLMs tend to model a large number of canceling interactions.

These findings help us gain deeper insights into the inference patterns of LLMs, particularly in high-stakes tasks where they provide quantitative metrics to indicate the degree to which LLM judgments can be trusted. The key contributions of the proposed mechanistic explanation method are,

• Revealing reasoning patterns, not just attribution. Attribution methods (48; 43; 32) tell us what words are important. For example, in "this movie is not bad," attribution highlights "not" and "bad." In contrast, the interaction-based approach (47; 26; 42) further reveals that the model relies on the interaction ("not", "bad") to reverse the sentiment. This allows us to distinguish whether the LLM is truly performing semantic composition or just using Bag-of-Words statistics.

• Evaluating the faithfulness and reliability of LLMs in high-stakes decisions. In high-stakes domains (e.g., medicine, finance, law), it is not enough for an LLM to be correct, it must be correct for the right reasons. For example, a legal LLM might learn a spurious correlation, associating a specific occupation with a "guilty" verdict (Case 2 in the paper). By quantifying interactions between input phrases, we can explicitly capture this biased reasoning. This provides a quantitative metric for reliability and serves as a tool for model auditing.

• Diagnosing shortcut learning. LLMs are adept at using statistical shortcuts to guess answers rather than performing genuine, complex reasoning. Interaction analysis can diagnose this behavior. For instance, when processing long texts, an LLM might rely only on a few local, low-order interactions to make a decision. By analyzing interaction order (complexity), we can quantify this phenomenon.

## F  MASKING IN EXPLAINABLE ARTIFICIAL INTELLIGENCE

Masking is a common practice in Explainable AI. For instance, in interaction-based methods (47; 26; 42), it is standard to evaluate many masked variants of input and restrict the analysis to short examples (*e.g.*, ≈10 phrases). Similarly, in perturbation-based attribution methods, such as LIME (43), Shapley sampling values, KernelSHAP (32), and DASP (2), it is also a widely adopted approach to evaluate numerous masked input variants, often restricting the analysis to short examples (*e.g.*, < 16 phrases).

Besides, this challenge can be alleviated through engineering techniques. First, we can use classic attribution methods (*e.g.* Integrated Gradients (48), LIME (43)) as a heuristic to identify and prioritize salient words or phrases, pruning the search space. Prior works (31; 27) have shown that LLMs exhibit attention on only a few sparse regions in inputs. Second, the input phrases in this paper are flexible units of analysis. They are not limited to single tokens but can represent multiple words, phrases, short sentences, or even paragraphs as input units (26; 42) Empirically, 10-12 input phrases were typically sufficient for effective analysis (43; 42; 11).

## G  HOW DOES THIS METHOD GUIDE MODEL IMPROVEMENT?

There are several methods that can be employed to enhance model performance.

**Enforce interaction consistency across models.** Reliable patterns are typically consistent across different models, while unreliable, high-order interactions are often model-specific and generalize poorly (26). We can jointly train two models using an interaction consistency loss to penalize differences in their learned patterns on the same input. This encourages models to converge on reliable reasoning, boosting overall performance.

**Refine supervised fine-tuning (SFT) with reliability scores.** We can integrate interaction analysis into both dataset construction and sample weighting. For dataset construction, identifying unreliable interactions allows us to create counterfactual data that explicitly targets weak reasoning spots, e.g., alleviating the identity discrimination. SFT instructions can also be designed to explicitly guide the model to use reliable interaction paths. For sample weighting, we can adjust sample weights based on the interaction reliability score ($s_{\text{reliable}}$). Samples where the model is correct but $s_{\text{reliable}}$ is very low (i.e., "correct output for the wrong reason") are assigned higher training weights, forcing the model to repair its underlying reasoning mechanism.

**Enhance reinforcement learning (RL) optimization.** We can incorporate interaction reliability into the reward dimension to shift RL optimization from the output result (What) to the reasoning process (How). For example, we can add the interaction reliability score ($s_{\text{reliable}}$) as a new feature to the reward model. The reward model would then reward generated texts not only based on human preference but also on highly reliable interaction paths. Besides, we can also impose constraints on the policy model during RL by applying an additional penalty term if the model's next token selection significantly increases the weight of unreliable interaction paths.

---

**Algorithm 1** Computing AND-OR interactions

---

1: **Input:** Input legal case $\mathbf{x}$, the legal LLM $v(\cdot)$, and the annotations of the relevant, irrelevant, and forbidden tokens in $\mathbf{x}$.
2: **Output:** A set of reliable interactions $I_{\text{and}}^{\text{reliable}}(S|\mathbf{x})$ and $I_{\text{or}}^{\text{reliable}}(S|\mathbf{x})$, and the ratio of reliable interaction effects $s^{\text{reliable}}$
3: Input the legal case $\mathbf{x}$ into the legal LLM, and generate the judgment (a sequence of tokens);
4: **for** $S \subseteq N$ **do**
5:     For each masked sample $\mathbf{x}_S$, compute the confidence score $v(\mathbf{x}_S)$ based on Eq. (1);
6: **end for**
7: **for** $S \subseteq N$ **do**
8:     Given $v(\mathbf{x}_S)$ for all combinations $S \subseteq N$, compute each AND interaction $I_S^{\text{AND}}$ and each OR interaction $I_S^{\text{OR}}$ via $\min_{\{\gamma_T\}} \sum_{S \subseteq N, S \neq \emptyset}[|I_S^{\text{AND}}| + |I_S^{\text{OR}}|]$;
9: **end for**
10: **for** $S \subseteq N$ **do**
11:     Compute the reliable AND interaction effect $I_{\text{and}}^{\text{reliable}}(S|\mathbf{x})$ and the reliable OR interaction effect $I_{\text{or}}^{\text{reliable}}(S|\mathbf{x})$ based on Eqs. (4) and (5).
12: **end for**
13: Compute the ratio of reliable interaction effects $s^{\text{reliable}}$ based on Eq. (7);
14: return $I_{\text{and}}^{\text{reliable}}(S|\mathbf{x})$, $I_{\text{or}}^{\text{reliable}}(S|\mathbf{x})$, $s^{\text{reliable}}$

---

## H OR INTERACTIONS CAN BE CONSIDERED SPECIFIC AND INTERACTIONS

The OR interaction $I_S^{\text{OR}}$ can be considered as a specific AND interaction $I_S^{\text{AND}}$, if we inverse the definition of the masked state and the unmasked state of an input variable.

Given a DNN $v : \mathbb{R}^n \to \mathbb{R}$ and an input sample $\mathbf{x} \in \mathbb{R}^n$, if we arbitrarily mask the input sample, we can get $2^n$ different masked samples $\mathbf{x}_S, \forall S \subseteq N$. Specifically, let us use baseline values $\mathbf{b} \in \mathbb{R}^n$ to represent the masked state of a masked sample $\mathbf{x}_S$, i.e.,

$$(\mathbf{x}_S)_i = \begin{cases} x_i, & i \in S \\ b_i, & i \notin S \end{cases} \tag{14}$$

Conversely, if we inverse the definition of the masked state and the unmasked state of an input variable, i.e., we consider $\mathbf{b}$ as the input sample, and consider the original value $\mathbf{x}$ as the masked

state, then the masked sample $\mathbf{b}_S$ can be defined as follows.

$$(\mathbf{b}_S)_i = \begin{cases} b_i, & i \in S \\ x_i, & i \notin S \end{cases} \tag{15}$$

According to the above definition of a masked sample in Equations (14) and (15), we can get $\mathbf{x}_{N \setminus S} = \mathbf{b}_S$. To simply the analysis, if we assume that $v_{\mathrm{and}}(\mathbf{x}_T) = v_{\mathrm{or}}(\mathbf{x}_T) = 0.5v(\mathbf{x}_T)$, then the OR interaction $I_S^{\mathrm{OR}}$ can be regarded as a specific AND interaction $I_S^{\mathrm{AND}}(\mathbf{b})$ as follows.

$$\begin{aligned} I_S^{\mathrm{OR}}(\mathbf{x}) &= -\sum\nolimits_{T \subseteq S} (-1)^{|S|-|T|} v_{\mathrm{or}}(\mathbf{x}_{N \setminus T}), \\ &= -\sum\nolimits_{T \subseteq S} (-1)^{|S|-|T|} v_{\mathrm{or}}(\mathbf{b}_T), \\ &= -\sum\nolimits_{T \subseteq S} (-1)^{|S|-|T|} v_{\mathrm{and}}(\mathbf{b}_T), \\ &= -I_S^{\mathrm{AND}}(\mathbf{b}). \end{aligned} \tag{16}$$

## I  ANNOTATION OF RELEVANT PHRASES, IRRELEVANT PHRASES, AND FORBIDDEN PHRASES

We propose the following **two principles** to avoid unnecessary ambiguity in the annotation of the three types of phrases. (1) The first principle is to avoid ambiguous legal cases. To ensure clarity, we engage several legal experts to select a set of straightforward and unambiguous legal cases. We let them to annotate the above three types of phrases to avoid ambiguity. (2) The second principle is to avoid analyzing subtle legal differences between the laws in different countries[7]. Although our algorithm can accurately explain the legal judgments made by legal LLMs based on sophisticated legal statutes, the goal of this paper is not to focus on such nuanced differences. Therefore, we let legal experts to select relatively simple and uncontroversial legal cases, enabling us to directly compare the performance of an English legal LLM and a Chinese legal LLM on the same input case.

For example, given an input legal case "*on June 1, during a conflict on the street, Andy stabbed Bob with a knife, causing Bob's death,*"[2] the legal LLM provides judgment "*murder*" for Andy. In above example, the input phrases can be set as $N = \{[on\ June\ 1], [during\ a\ conflict], [on\ the\ street], [Andy\ stabbed\ Bob\ with\ a\ knife], [causing\ Bob's\ death]\}$. $\mathcal{R} = \{[Andy\ stabbed\ Bob\ with\ a\ knife], [causing\ Bob's\ death]\}$ are the direct reason for the judgment, thereby being annotated as *relevant phrases*, where all tokens in the brackets [] are taken as a single input phrase. The set of irrelevant phrases are annotated as $\mathcal{I} = \{[on\ June\ 1], [during\ a\ conflict], [on\ the\ street]\}$. The input phrase like "*during a conflict*" may influence Andy's behavior "*Andy stabbed Bob with a knife,*" but it is the input phrase "*Andy stabbed Bob with a knife*" that directly contributes to the legal judgment of "*murder,*" rather than the input phrase "*during a conflict.*"

Given another input legal case involving multiple individuals, such as "*Andy assaulted Bob on the head, causing minor injuries. Charlie stabbed Bob with a knife, causing Bob's death,*"[2] the legal LLM assigns the judgment of "*assault*" to Andy.

Let the set of all input phrases be $N = \{[Andy\ assaulted\ Bob\ on\ thehead], [causing\ minor\ injuries], [Charlie\ stabbed\ Bobwith\ a\ knife], [causing\ Bob's\ death]\}$. Although the input phrases "*Charlie stabbed Bob with a knife*" and "*causing Bob's death*" naturally all represent crucial facts for judgment, they should not influence the judgment for Andy, because these words describe the actions of Charlie, not actions of Andy. Therefore, these input phrases are annotated as forbidden phrases, $\mathcal{F} = \{[Charlie\ stabbed\ Bob\ with\ a\ knife], [causing\ Bob's\ death]\}$.

## J  FAITHFULNESS OF THE INTERACTION-BASED EXPLANATION

In this section, we conducted experiments to evaluate the **sparsity property** in Figure 7 and the **universal matching property** in Figure 8 of the extracted interactions.

---

[7]We use an English legal LLM SaulLM-7B-Instruct (7), which is trained using legal corpora from English-speaking jurisdictions such as the U.S., Canada, the UK, and Europe, and we use a Chinese legal LLM BAI-Law-13B (21), which is trained using legal corpora from China.

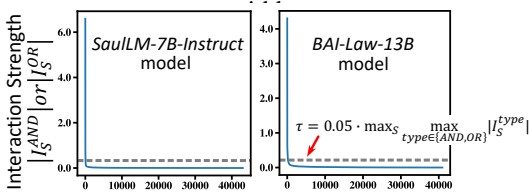

Figure 7: Sparsity property of interactions. We show the strength of different AND-OR interactions ($|I_S^{\text{AND}}|$ and $|I_S^{\text{OR}}|$) extracted from different samples in a descending order. Only about 0.5% interactions had salient effects.

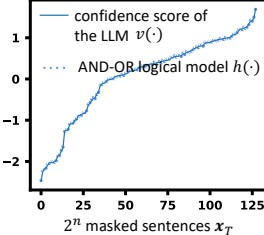

Figure 8: Universal matching property of interactions. Experiment verifies that the surrogate logical model $h(\mathbf{x}_T)$ can accurately fit the confidence scores of the LLM $v(\mathbf{x}_T)$ on all $2^n$ masked samples $\{\mathbf{x}_T | T \subseteq N\}$, i.e., $\forall T \subseteq N, v(\mathbf{x}_T) = h(\mathbf{x}_T)$, no matter how we randomly mask the input sample $\mathbf{x}$ in $2^n$ different masking states $T \subseteq N$.

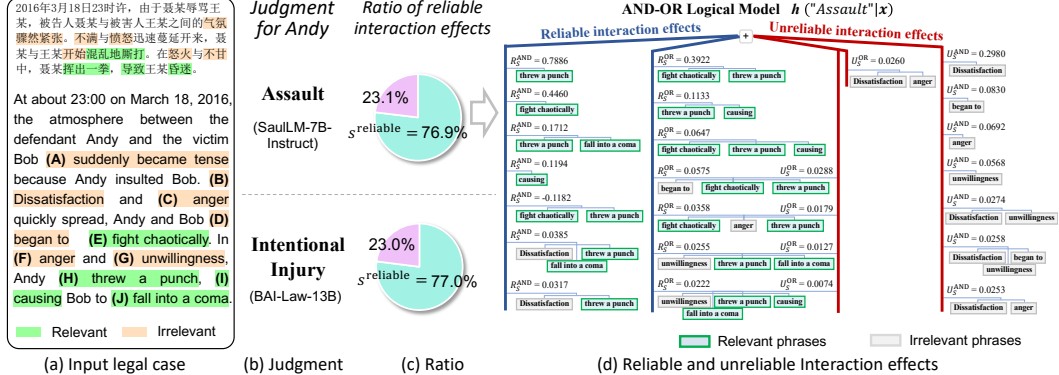

Figure 9: Visualization of judgments influenced by unreliable irrelevant phrases. (a) Irrelevant phrases include sentimental phrases that are not the direct reason for judgment. Criminal actions were annotated as relevant phrases. We also translated the legal case to English as the input of the SaulLM-7B-Instruct model. (b) Judgments predicted by the two legal LLMs, which were both correct according to laws of the two countries. (c,d) We quantified the reliable and unreliable interaction effects.

## K  MAKING JUDGMENTS BASED ON SEMANTICALLY IRRELEVANT PHRASES

**Case 3: making judgments based on unreliable irrelevant phrases.** We observed that although legal LLMs achieved great performance in predicting legal judgment results, the legal LLMs used a significant portion of interaction patterns that were attributed to semantically irrelevant phrases for judgment (e.g., the time, the location, and the sentimental phrases that are not the direct reason for the judgment). To evaluate the impact of semantically irrelevant phrases on both the SaulLM-7B-Instruct and BAI-Law-13B models, we engaged legal experts to annotate phrases that served as the direct reason for the judgment as relevant phrases in $\mathcal{R}$, and those that were not the direct reason for the judgment as irrelevant phrases in $\mathcal{I}$, e.g., semantically irrelevant phrases and unreliable sentimental phrases behind real criminal actions.

Figure 9 shows the first legal case, which showed Andy had a conflict with Bob and attacked Bob, committing an assault. Here, input phrases such as "*fight chaotically,*" "*threw a punch,*" "*causing,*" and "*fall into a coma*" were annotated as relevant phrases in $\mathcal{R}$, as these phrases served as the

direct reason for the judgment "*Assault*." On the other hand, input phrases like "*began to*" and sentiment-driven phrases such as "*dissatisfaction*," "*anger*" were annotated as irrelevant phrases in $\mathcal{I}$, as these phrases were not direct reason for the judgment.

In this legal case, there were 28 AND interaction patterns and 22 OR interaction patterns in the top 50 most salient AND-OR interaction patterns. Here, the average interaction strength for top 50 most salient interactions was 0.078, while the average interaction strength for the remaining AND-OR interaction patterns among the $2 \times 2^{10} = 2048$ AND-OR interaction patterns was 0.005. The legal LLM SaulLM-7B-Instruct *did* use several interaction patterns that aligned with legal experts' domain knowledge for the legal judgment. For example, an AND interaction pattern $S_1 = \{$"*threw a punch*"$\}$, and an AND interaction pattern $S_2 = \{$"*threw a punch*", "*fall into a coma*"$\}$, and an OR interaction pattern $S_3 = \{$"*fight chaotically*", "*threw a punch*"$\}$ contributed salient reliable interaction effects $R_{S_1}^{\text{AND}} = 0.79$, $R_{S_2}^{\text{AND}} = 0.17$, and $R_{S_3}^{\text{OR}} = 0.39$ to the confidence score $v($"*Assault*"$|\mathbf{x})$ of the judgment "*Assault*," respectively. However, the legal LLM also used lots of interaction patterns that did not match legal experts' domain knowledge for the legal judgment. For example, two AND interaction patterns $S_4 = \{$"*dissatisfaction*"$\}$, and $S_5 = \{$"*anger*"$\}$, which represented unreliable sentiments instead of criminal actions, contributed salient unreliable interaction effects $U_{S_4}^{\text{AND}} = 0.30$ and $U_{S_5}^{\text{AND}} = 0.07$ to the confidence score of the judgment "*Assault*," respectively. In sum, the SaulLM-7B-Instruct model used a ratio of $s^{\text{reliable}} = 76.9\%$ reliable interaction effects for the legal judgment. This indicated that the legal LLM mistakenly made judgments based on unreliable irrelevant phrases, because unreliable sentimental tokens only served as explanations for criminal actions, rather than the direct reason for the legal judgments.

In comparison, we evaluated the above legal case on the BAI-Law-13B model, as shown in Figure 9 and Figure 10 in Appendix. There were 12 AND interaction patterns and 38 OR interaction patterns in the top 50 most salient AND-OR interaction patterns. The average interaction value for top 50 most salient interactions was 0.048, while the average interaction value for the remaining AND-OR interaction patterns was 0.004. Compared to the SaulLM-7B-Instruct model's $s^{\text{reliable}} = 76.9\%$ ratio of reliable interaction effects, the BAI-Law-13B model used similar reliable interactions, *i.e.*, using a ratio of $s^{\text{reliable}} = 77.0\%$ reliable interaction effects and a ratio of $s^{\text{unreliable}} = 23.0\%$ unreliable interaction effects to compute the confidence score. Many interaction patterns used by the BAI-Law-13B model were also used by the SaulLM-7B-Instruct model, such as an AND interaction pattern $S_1 = \{$"*threw a punch*"$\}$, and an OR interaction pattern $S_2 = \{$"*fight chaotically*", "*threw a punch*"$\}$ contributed salient reliable interaction effects $R_{S_1}^{\text{AND}} = 0.34$ and $R_{S_2}^{\text{OR}} = 0.12$ to the confidence score $v($"*Intentional Injury*"$|\mathbf{x})$ of the judgment "*Intentional Injury*," respectively. This indicated that these two legal LLMs did successfully identify some direct reasons for the legal judgment. On the other hand, the BAI-Law-13B model used problematic interaction patterns for the legal judgment, such as two AND interaction patterns $S_3 = \{$"*suddenly became tense*"$\}$ and $S_4 = \{$"*anger*"$\}$ contributed salient unreliable interaction effects $U_{S_3}^{\text{AND}} = 0.08$ and $U_{S_4}^{\text{AND}} = 0.03$ to the confidence score, respectively. The unreliable sentimental token should not be used to determine the judgment. Additional examples of making judgments based on unreliable sentimental phrases are provided make judgment on Andy in Section L.3.

## L   MORE EXPERIMENT RESULTS AND DETAILS

### L.1   DISTRIBUTION OF PHRASE ANNOTATIONS BY LEGAL EXPERTS AND VOLUNTEERS

In this subsection, we show the distribution of phrase annotations provided by legal experts. Specifically, we consulted 16 legal experts to annotate the phrases in the input prompts using a majority voting scheme. The selected cases are generally simple and straightforward, ensuring that phrase annotations are relatively clear and unlikely to introduce major issues.

**Legal background of legal experts.** These legal experts are either working in the legal profession or studying law-related disciplines. Their experience in the legal field ranges from two to twelve years, with academic backgrounds in areas such as criminal procedure law, international law, and jurisprudence. Specifically, three of these experts have over eight years of experience as criminal trial judges, one serves as an assistant to a criminal trial judge, and two are currently pursuing master degrees in international law. The diverse backgrounds of these legal professionals greatly contribute

Table 1: Phrase annotation for Case 1.

| Input phrase | Is relevant phrase? | Is irrelevant phrase? | Is forbidden phrase? | Final annotation |
|---|---|---|---|---|
| (A) tense | 0 | 16 | 0 | Irrelevant phrase |
| (B) Dissatisfaction | 0 | 16 | 0 | Irrelevant phrase |
| (C) anger | 0 | 16 | 0 | Irrelevant phrase |
| (D) began to | 2 | 14 | 0 | Irrelevant phrase |
| (E) fight chaotically | 16 | 0 | 0 | Relevant phrase |
| (F) anger | 3 | 13 | 0 | Irrelevant phrase |
| (G) unwillingness | 3 | 13 | 0 | Irrelevant phrase |
| (H) threw a punch | 16 | 0 | 0 | Relevant phrase |
| (I) causing | 16 | 0 | 0 | Relevant phrase |
| (J) fall into a coma | 16 | 0 | 0 | Relevant phrase |

Table 2: Phrase annotation for Case 2.

| Input phrase | Is relevant phrase? | Is irrelevant phrase? | Is forbidden phrase? | Final annotation |
|---|---|---|---|---|
| (A) morning | 0 | 16 | 0 | Irrelevant phrase |
| (B) had an argument | 3 | 13 | 0 | Irrelevant phrase |
| (C) chased | 14 | 2 | 0 | Relevant phrase |
| (D) with an axe | 15 | 1 | 0 | Relevant phrase |
| (E) bit | 15 | 1 | 0 | Relevant phrase |
| (F) slightly injured | 16 | 0 | 0 | Relevant phrase |
| (G) hit | 3 | 0 | 13 | Forbidden phrase |
| (H) with a shovel | 0 | 0 | 16 | Forbidden phrase |
| (I) injuring | 0 | 0 | 16 | Forbidden phrase |
| (J) death | 1 | 0 | 15 | Forbidden phrase |

to the analysis of relevant, irrelevant, and forbidden phrases in legal cases, providing a nuanced legal perspective.

**Distribution of phrase annotations.** We present the distribution of phrase annotations for each phrase in the three legal cases discussed in the main paper, as shown in Table 1, Table 2 and Table 3. The final annotation for each phrase in the input legal case was determined using a majority voting scheme.

**Case 1**: At about 23:00 on March 18, 2016, the atmosphere between the defendant Andy and the victim Bob suddenly became tense because Andy insulted Bob. Dissatisfaction and anger quickly spread, and Andy and Bob began to fight chaotically. In anger and unwillingness, Andy threw a punch, causing Bob to fall into a coma.

Judgment of the legal LLM for Andy: Assault.

Table 3: Phrase annotation for Case 3.

| Input phrase | Is relevant phrase? | Is irrelevant phrase? | Is forbidden phrase? | Final annotation |
|---|---|---|---|---|
| (A) Wednesday | 0 | 16 | 0 | Irrelevant phrase |
| (B) night | 0 | 16 | 0 | Irrelevant phrase |
| (C) a judge | 0 | 16 | 0 | Irrelevant phrase |
| (D) walked | 0 | 16 | 0 | Irrelevant phrase |
| (E) home | 0 | 16 | 0 | Irrelevant phrase |
| (F) a day's work | 0 | 16 | 0 | Irrelevant phrase |
| (G) dark | 0 | 16 | 0 | Irrelevant phrase |
| (H) holding a knife | 16 | 0 | 0 | Relevant phrase |
| (I) robbed | 16 | 0 | 0 | Relevant phrase |
| (J) belongings | 16 | 0 | 0 | Relevant phrase |

**Case 2**: On the morning of December 22, 2013, the defendants Andy and Bob deceived Charlie and the three of them had an argument. Andy chased Charlie with an axe and bit Charlie, causing Charlie to be slightly injured. Bob hit Charlie with a shovel, injuring Charlie and causing Charlie' death.

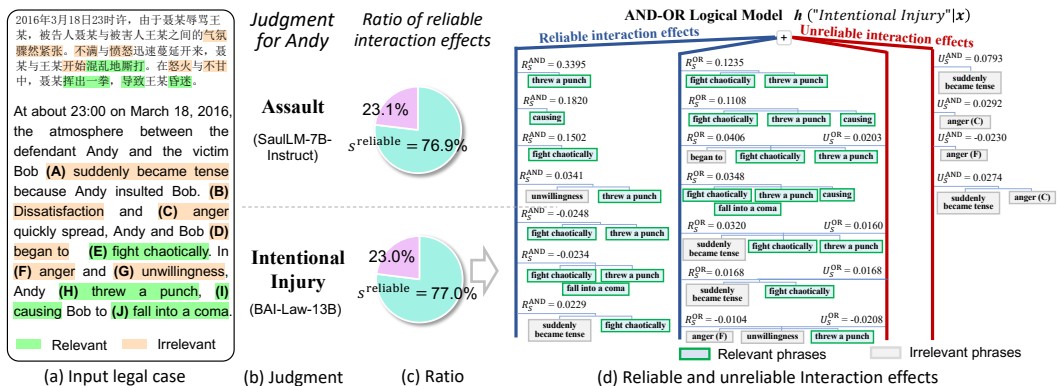

Figure 10: Visualization of judgments influenced by unreliable irrelevant phrases in the BAI-Law-13B model. (a) Irrelevant phrases include sentimental phrases that are not the direct reason for judgment. Criminal actions were annotated as relevant phrases. We also translated the legal case to English as the input of the SaulLM-7B-Instruct model. (b) Judgments predicted by the two legal LLMs, which were both correct according to laws of the two countries. (c,d) We quantified the reliable and unreliable interaction effects.

Judgment of the legal LLM for Andy: Assault.

**Case 3**: Late Wednesday night, Andy, a judge, walked home alone after finishing a day's work. On the dark road, two suspicious men followed, holding a knife and robbed Andy's belongings.

Judgment of the legal LLM for two suspicious men: Robbery.

## L.2 SIGNIFICANCE OF CONFLICTED INTERACTION PATTERNS

This subsection shows the significance of mutual cancellation of interaction patterns. We found that over 60% effects of the interaction patterns had been mutually cancelled out in Table 4.

Table 4: Significance of mutual cancellation of interaction patterns (%), which is measured by $s^{\text{conflict}}$.

| Dateset | Qwen | Deepseek | BAI | SaulLM |
|---|---|---|---|---|
| CAIL2018 | 78.00 | 82.46 | 62.67 | - |
| LeCaRD | 78.70 | 65.58 | 85.38 | - |
| LEVEN | 78.40 | 77.58 | 83.72 | - |
| LegalBench | 75.31 | 83.10 | - | 76.60 |
| LexGLUE | 98.25 | 96.18 | - | 29.97 |

## L.3 MORE RESULTS OF JUDGMENTS INFLUENCED BY UNRELIABLE SENTIMENTAL TOKENS

We conducted more experiments to show the judgments influenced by unreliable sentimental tokens in Figure 12, Figure 13, and Figure 14, respectively. We observed that a considerable number of interactions contributing to the confidence score $v(\mathbf{x})$ were attributed to semantically irrelevant or unreliable sentimental tokens. In different legal cases, the ratio of reliable interaction effects to all salient interactions was within the range of 32.6% to 87.1%. It means that about 13~68% of interactions used semantically irrelevant tokens or unreliable sentimental tokens for the judgment.

## L.4 MORE RESULTS OF JUDGMENTS AFFECTED BY INCORRECT ENTITY MATCHING

We conducted more experiments to show the judgments affected by incorrect entity matching in Figure 15, Figure 16, and Figure 17, respectively. We observed that a considerable ratio of the confidence score $v(\mathbf{x})$ was mistakenly attributed to interactions on criminal actions made by incorrect entities. In different legal cases, the ratio of reliable interaction effects to all salient interactions was within the range of 31.9% to 67.8%. It means that about 22~68% of interactions used semantically

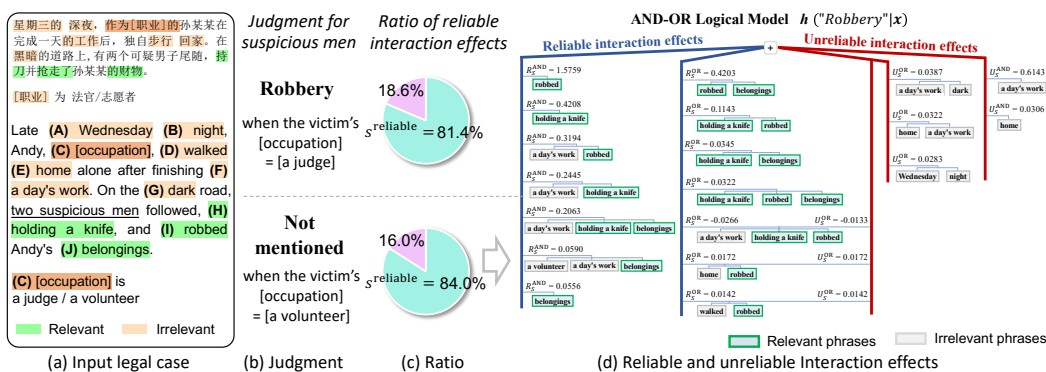

Figure 11: Visualization of judgments biased by discrimination in identity, when the victim's [occupation] is [a volunteer]. To enable the fair comparison, we compute interactions on the output score $v(\text{"}Robbery\text{"}|\mathbf{x})$, instead of the actual LLM's output score $v(\text{"}Not\ mentioned\text{"}|\mathbf{x})$. (a) Irrelevant phrases were annotated in the legal case, including the occupation, time and actions that are not the direct reason for the judgment. Criminal actions of the defendant were annotated as relevant phrases. (b) The SaulLM-7B-Instruct model predicted the judgment based on the legal case with different occupations. (c,d) We quantified the reliable and unreliable interaction effects.

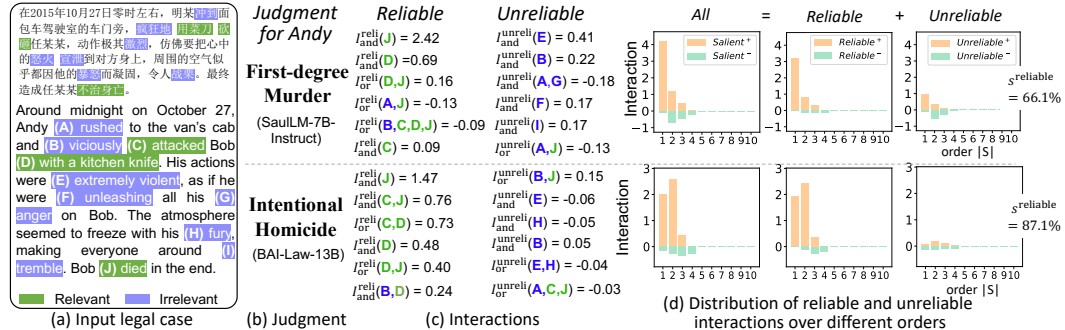

Figure 12: More results of judgments influenced by unreliable sentimental tokens. (a) A number of irrelevant tokens were annotated in the legal case, including unreliable sentimental tokens. Criminal actions were annotated as relevant tokens. We also translated the legal case to English as the input of the SaulLM-7B-Instruct model. (b) Judgements predicted by the two legal LLMs, which were both correct according to laws of the two countries. (c,d) We quantified the reliable and unreliable interaction effects of different orders. The SaulLM-7B-Instruct model used 66.1% reliable interaction effects, while the BAI-Law-13B model encoded 87.2% reliable interaction effects.

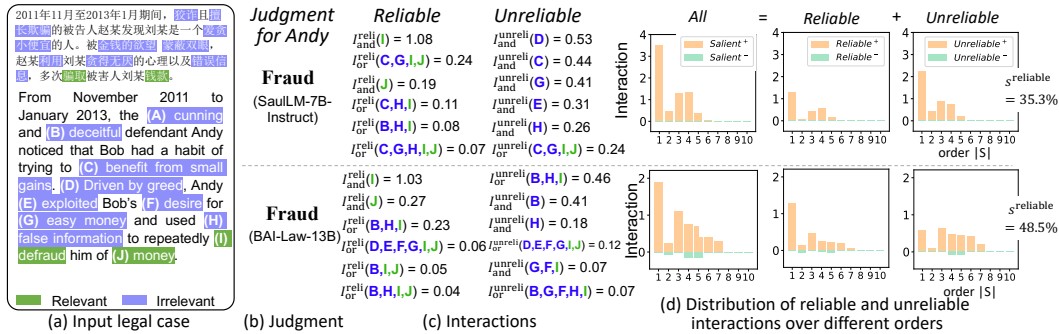

Figure 13: More results of judgments influenced by unreliable sentimental tokens. (d) The SaulLM-7B-Instruct model used 35.3% reliable interaction effects, while the BAI-Law-13B model encoded 48.5% reliable interaction effects.

irrelevant tokens for the judgment, or was mistakenly attributed on criminal actions made by incorrect entities.

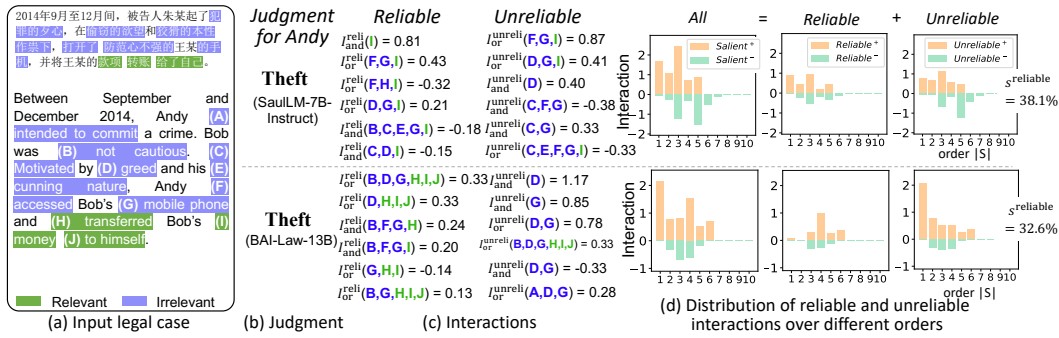

Figure 14: More results of judgments influenced by unreliable sentimental tokens. (d) The SaulLM-7B-Instruct model used 38.1% reliable interaction effects, while the BAI-Law-13B model encoded 32.6% reliable interaction effects.

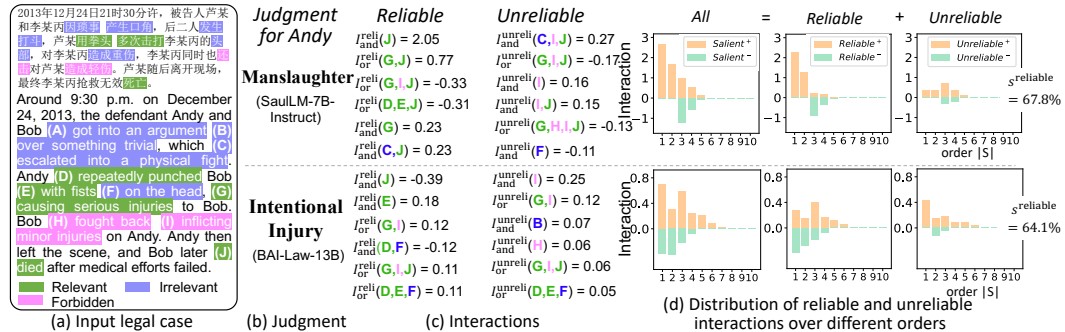

Figure 15: More results of judgments affected by incorrect entity matching. (a) A number of irrelevant tokens were annotated in the legal case, including the time and actions that were not the direct reason for the judgment. Criminal actions of the defendant were annotated as relevant tokens. Criminal actions of the unrelated person were annotated as forbidden tokens. (b) Judgements predicted by the two legal LLMs, which were both correct according to laws of the two countries. (c,d) We measured the reliable and unreliable interaction effects of different orders. The SaulLM-7B-Instruct model used 67.8% reliable interaction effects, while the BAI-Law-13B model encoded 64.1% reliable interaction effects.

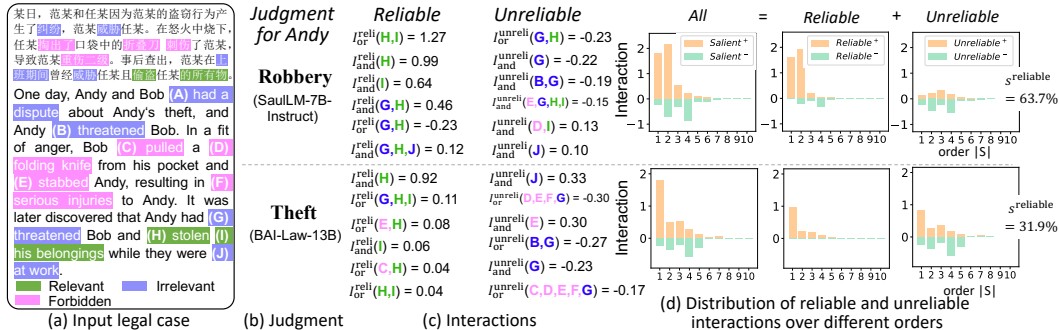

Figure 16: More results of judgments affected by incorrect entity matching. (d) The SaulLM-7B-Instruct model used 63.7% reliable interaction effects, while the BAI-Law-13B model encoded 31.9% reliable interaction effects.

## L.5    MORE RESULTS OF JUDGMENTS BIASED BY DISCRIMINATION IN OCCUPATION

**Experiment results of judgments biased by discrimination in occupation in Section 3.** Figure 21 illustrates additional examples of how occupation influences the judgment of the legal case, which were tested on the SaulLM-7B-Instruct model. It shows that if we replaced "*a judge*" with law-related occupations, such as "*a lawyer*" and "*a policeman,*" the judgment remained "*robbery.*" Besides, the occupation "*a programmer*" changed the judgment to "*not mentioned.*" The interactions containing

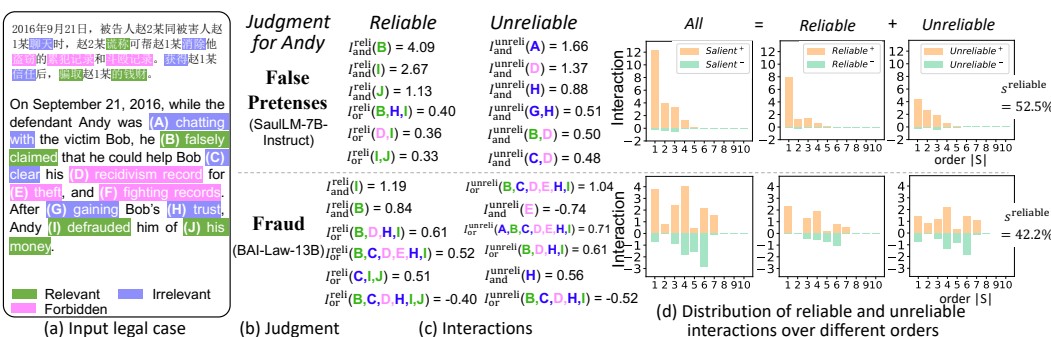

Figure 17: More results of judgments affected by incorrect entity matching. (d) The SaulLM-7B-Instruct model used 52.5% reliable interaction effects, while the BAI-Law-13B model encoded 42.2% reliable interaction effects.

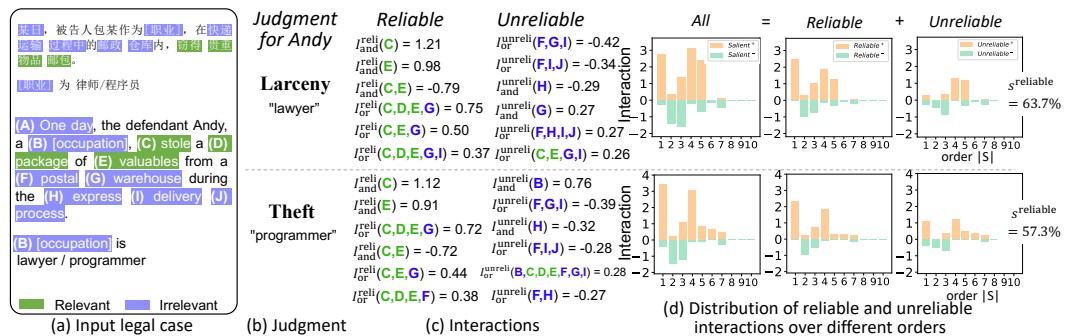

Figure 18: More results of judgments biased by discrimination in occupation. (a) A number of irrelevant tokens were annotated in the legal case, including the occupation, time and actions that are not the direct reason for the judgment. Criminal actions of the defendant were annotated as relevant tokens. (b) The SaulLM-7B-Instruct model predicted the judgment based on the legal case with different occupations, respectively. (c,d) We measured the reliable and unreliable interaction effects of different orders. When the occupation was set to "*lawyer*," the LLM used 63.7% reliable interaction effects. In comparison, when the occupation was set to "*programmer*," the LLM encoded 57.3% reliable interaction effects.

the occupation token (*i.e.*, "*a judge*", "*a lawyer*", "*a policeman*", "*a programmer*", and "*a volunteer*") were important factors that changed the ratio of reliable interactions from 81.4% to 84.0%. This suggested that the legal LLM sometimes had considerable occupation bias.

Futhermore, Figure 22 shows the test of the BAI-Law-13B model on the legal case, in which *Andy, the victim with varying occupations, was robbed of his belongings by two suspicious men*. Similarly, we found that the BAI-Law-13B model encoded interactions with the occupation tokens "*a judge*," which boosted the confidence of the judgment "*robbery*." More interestingly, if we substituted the occupation tokens "*a judge*" to "*a policeman*," the interaction of the occupation "*a policeman*," decreased from 0.29 to 0.11. The interactions containing the occupation token were important factors that changed the ratio of reliable interactions from 78.9% to 87.1%. This suggested that the legal LLM sometimes had considerable occupation bias.

**More results of judgments biased by discrimination in occupation.** We conducted more experiments to show the judgments biased by discrimination in occupation in Figure 18, Figure 19, and Figure 20, respectively. We found that the legal LLM usually used interactions on the occupation information to compute the confidence score $v(\mathbf{x})$. In different legal cases, the ratio of reliable interaction effects to all salient interactions was within the range of 30.1% to 63.7%. In particular, in Figure 18, changing the occupation from "*lawyer*" to "*programmer*" results in a decrease of the reliable interactions from 63.7% to 57.3%. The difference of interactions containing the occupation token changes the model output from "*Larceny*" to "*Theft*."

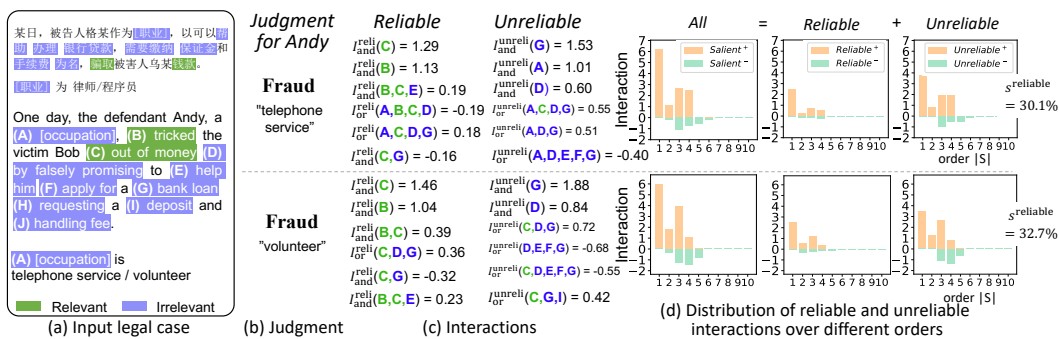

Figure 19: More results of judgments biased by discrimination in occupation. (b) The SaulLM-7B-Instruct model predicted the judgment based on the legal case with different occupations, respectively. (d) When the occupation was set to "*telephone service*," the LLM used 30.1% reliable interaction effects. In comparison, when the occupation was set to "*volunteer*," the LLM encoded 32.7% reliable interaction effects.

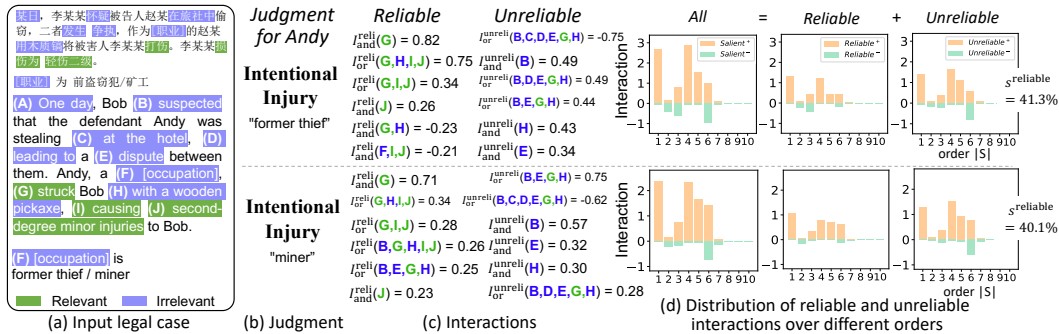

Figure 20: More results of judgments biased by discrimination in occupation. (b) The BAI-Law-13B model predicted the judgment based on the legal case with different occupations, respectively. (d) When the occupation was set to "*former thief*," the LLM used 41.3% reliable interaction effects. In comparison, when the occupation was set to "*miner*," the LLM encoded 40.1% reliable interaction effects.

### L.6 EXPERIMENT DETAILS OF MASKED SAMPLES

This section discusses how to obtain the masked sample $\mathbf{x}_T, T \subseteq N$. Given the confidence score of a DNN $v(\mathbf{x})$ and an input sample $\mathbf{x} = [x_1, x_2, \cdots, x_n]^\intercal$ with $n$ input phrases, if we arbitrarily mask the input sample $\mathbf{x}$, we can get $2^n$ different masked samples $\mathbf{x}_T, \forall T \subseteq N$. Specifically, for each input variable $i \in N \setminus T$, we replace it with the baseline value $b_i$ to represent its masked state. Let us use baseline values $\mathbf{b} = [b_1, b_2, \cdots, b_n]^\intercal$ to represent the masked state of a masked sample $\mathbf{x}_T$, *i.e.*,

$$(\mathbf{x}_T)_i = \begin{cases} x_i, & i \in T \\ b_i, & i \notin T \end{cases} \tag{17}$$

For sentences in a language generation task, the masking of input phrases is performed at the embedding level. Following the approach of (42; 46), we masked inputs at the embedding level by transforming sentence tokens into their corresponding embeddings. Given an input sentence $\mathbf{x} = [x_1, x_2, \cdots, x_n]^\intercal$ with $n$ input tokens, the $i$-th token $x_i$ is mapped to its embedding $e_i \in \mathbb{R}^d$, where $d$ is the dimension of the embedding layer. To obtain the masked sample $\mathbf{x}_T$, if $i \in N \setminus T$, the embedding is replaced with the (constant) baseline value $b_i \in \mathbb{R}^d$, *i.e.*, $e_i = b_i$. Otherwise, the embedding remains unchanged, *i.e.*, $e_i = e_i$. Following (40), we trained the (constant) baseline value $b_i \in \mathbb{R}^d$ to extract the sparsest interactions.

### L.7 EXPERIMENT DETAILS OF USING THE SAME DATASET FOR COMPARISON

This section presents the experiment details of using the CAIL2018 dataset (56) to ensure a fair comparison between two legal LLMs. For the BAI-Law-13B model, a Chinese legal LLM, we directly

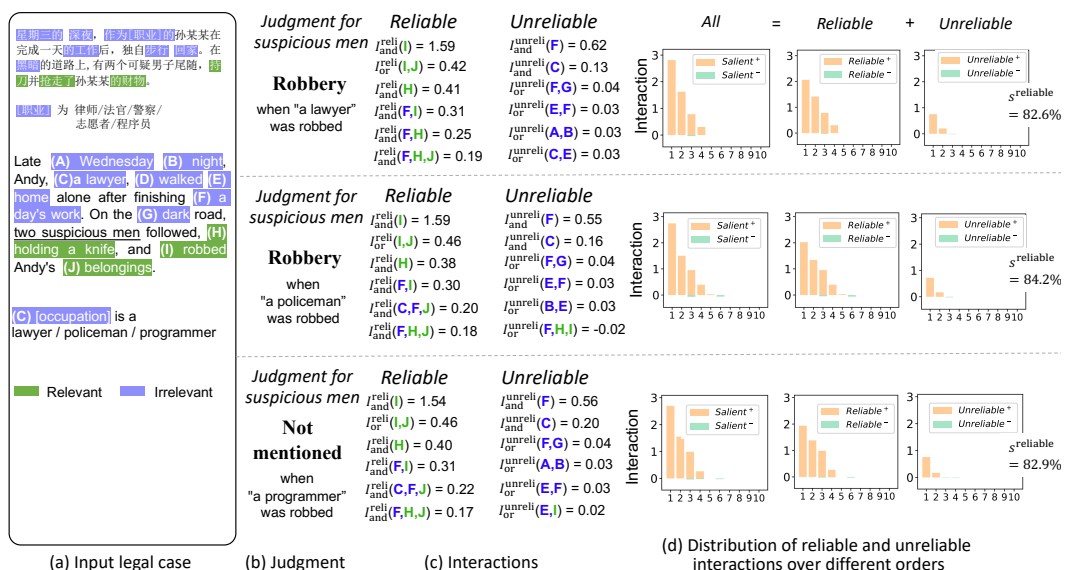

(a) Input legal case     (b) Judgment     (c) Interactions     (d) Distribution of reliable and unreliable interactions over different orders

Figure 21: Visualization of judgments biased by discrimination in occupation. (a) A number of irrelevant tokens were annotated in the legal case, including the occupation, time and actions that are not the direct reason for the judgment. Criminal actions of the defendant were annotated as relevant tokens. (b) The SaulLM-7B-Instruct model predicted the judgment based on the legal case with different occupations, respectively. (c,d) We measured the reliable and unreliable interaction effects of different orders. When the occupation was set to "*a lawyer*," the LLM used 82.6% reliable interaction effects. In comparison, when the occupation was set to "*a policeman*," the LLM encoded 84.2% reliable interaction effects.

analyzed the Chinese legal cases from the CAIL2018 dataset. In contrast, for the SaulLM-7B-Instruct model, an English legal LLM, we translated the Chinese legal cases into English and performed the analysis on the translated cases, to enable fair comparisons. To simplify the explanation and avoid ambiguity, we only explained the inference patterns on legal cases, which were correctly judged by the LLM.

Starting with a complete fact descriptions of the legal case from the CAIL2018 dataset, we first condensed the case by removing descriptive details irrelevant to the judgment, retaining only the most informative tokens, such as the time, location, people, and events. To prompt the model to deliver its judgment, we added a structured prompt designed to extract a concise answer. The format is as follows:

"*Question: [Fact descriptions of the case]. What crime did [the defendant] commit? Briefly answer the specific charge in one word. Answer: The specific charge is*"

Here, *[Fact descriptions of the case]* is replaced with the details of the specific legal case, and *[the defendant]* is substituted with the name of the defendant.

To identify potential representation flaws behind the seemingly correct language generation results of legal LLMs, we introduced special tokens that were irrelevant to the judgments. For cases to assess if judgments were influenced by unreliable sentimental tokens, we added such tokens to describe actions in the legal case. We then observed whether a substantial portion of the interactions contributing to the confidence score $v(\mathbf{x})$ were associated with semantically irrelevant or unreliable sentimental tokens. Similarly, in cases where we aimed to detect potential bias based on occupation, we included irrelevant occupation-related tokens for the defendants or victims, and analyzed whether the legal LLM leveraged these occupation-related tokens to compute the confidence score $v(\mathbf{x})$ in Eq. (1).

Finally, we show the selection of input phrases for extracting interactions. As discussed in Section 2.1, given an input sample $\mathbf{x}$ with $n$ input phrases, we can extracted at most $2^{n+1}$ AND-OR interactions to compute the confidence score $v(\mathbf{x})$. Consequently, the computational cost for extracting interactions increases exponentially with the number of input phrases. To alleviate this issue, we followed (42; 46) to select a set of tokens as input phrases, while keeping the remaining tokens as a constant background

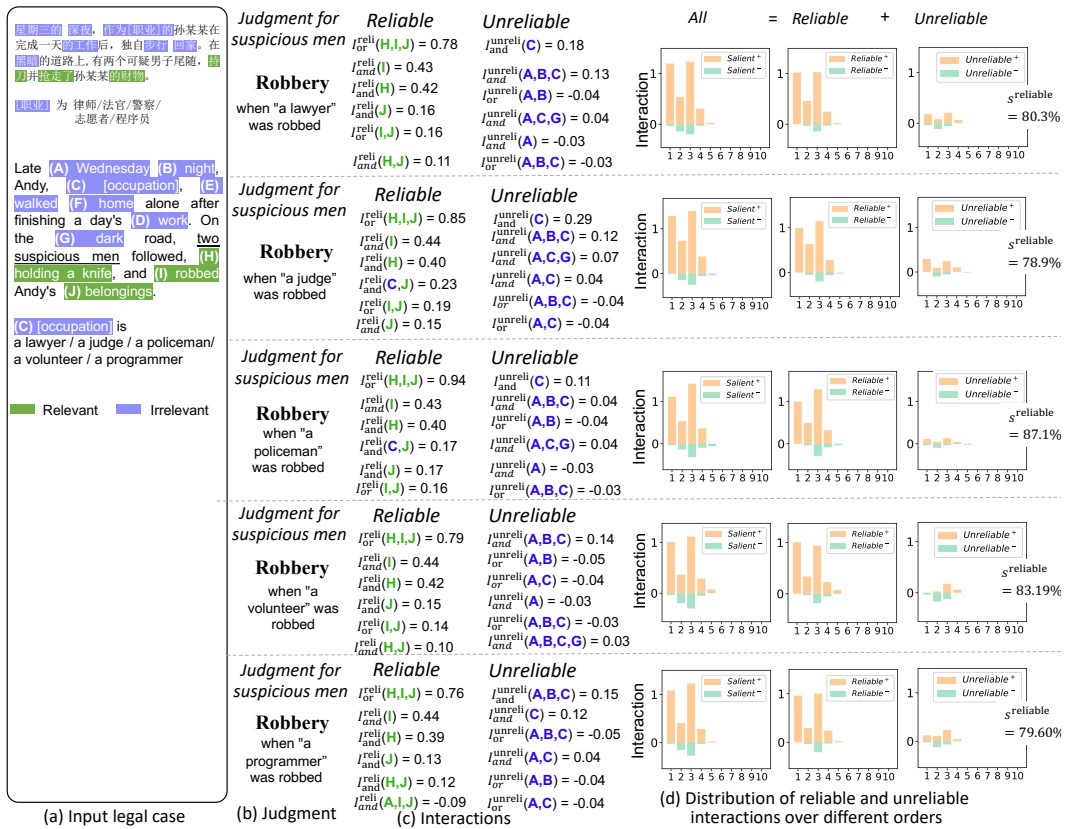

Figure 22: Visualization of judgments biased by discrimination in occupation. (a) A number of irrelevant tokens were annotated in the legal case, including the occupation, time and actions that are not the direct reason for the judgment. Criminal actions of the defendant were annotated as relevant tokens. (b) The BAI-Law-13B model predicted the judgment based on the legal case with different occupations, respectively. (c,d) We measured the reliable and unreliable interaction effects of different orders. When the occupation was set to "*a judge*," the LLM used 78.9% reliable interaction effects. In comparison, when the occupation was set to "*a policeman*," the LLM encoded 87.1% reliable interaction effects.

in Section L.6, to compute interactions among the selected variables. Specifically, we selected 10 informative input phrases (tokens or phrases) for each legal case. These input phrases were manually selected based on their informativeness for judgements. It was ensured that the removal of all input phrases would substantially change the legal judgment result.

## L.8 MORE APPLICATIONS OF THE METHOD

While this paper focuses on the legal domain due to space constraints, the proposed method is generic. It can be applied to quantify the decision trustworthiness of any LLM, particularly in high-stakes domains such as medicine and finance. In medicine, the method can help doctors understand which aspects of a patient's record the LLM is focusing on for diagnosis, allowing them to assess the reliability of its reasoning. Similarly, in finance, it can clarify an LLM's credit-scoring logic, revealing which applicant features the model prioritizes and the reliability of that reasoning.

Furthermore, the applicability of our method extends from natural language processing to computer vision. Specifically, the proposed method can be used for pedestrian detection analysis to determine whether the model relies on unreliable inference patterns for its model inference.

Figure 23 shows the interactions extracted from a trained DNN for pedestrian detection. Given an input image, we first manually label image regions with salient attributions as input variables, and

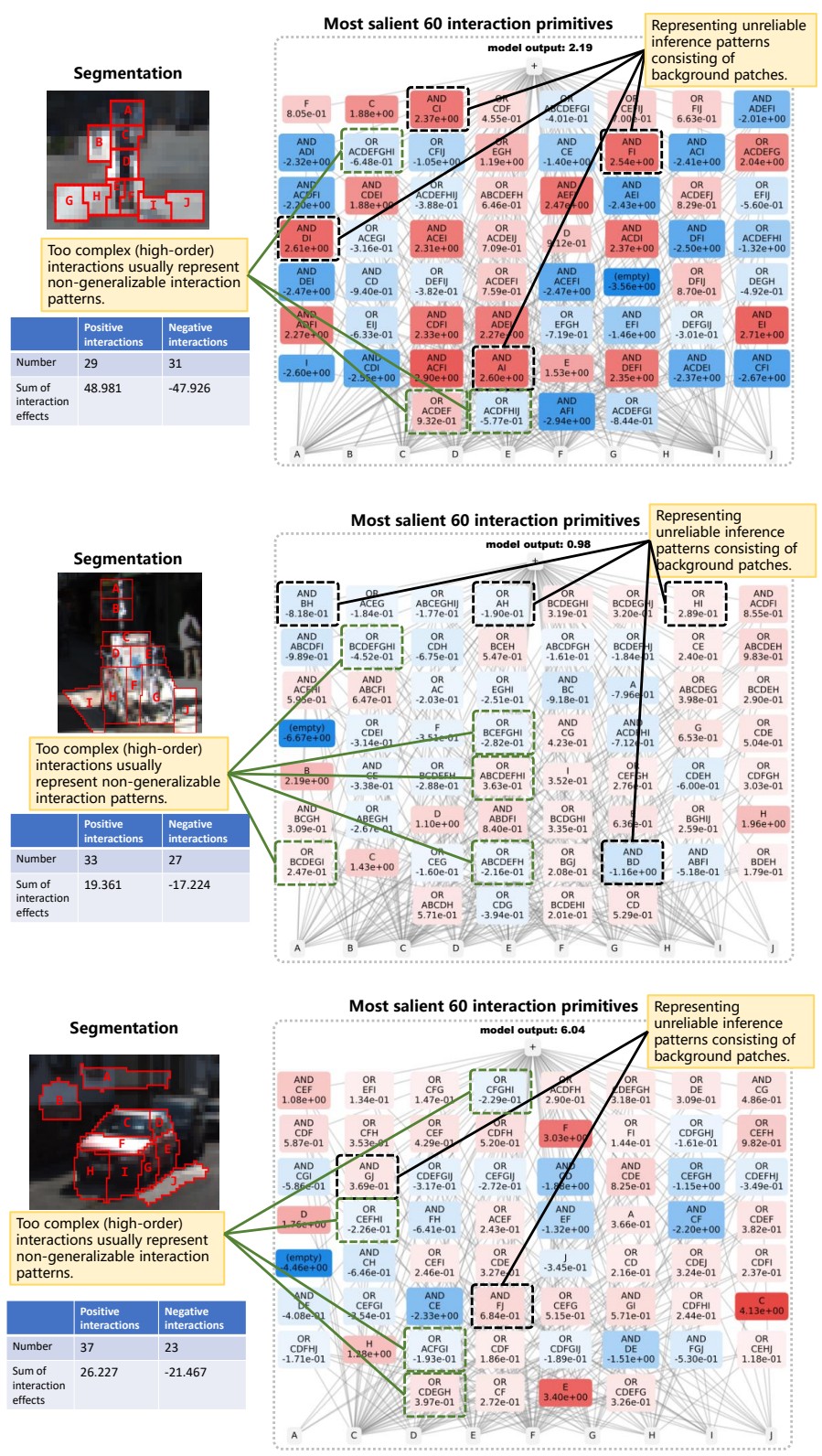

Figure 23: The interactions extracted from a DNN for pedestrian detection.

then compute interactions between these image regions. The visualization of the interactions enables human to manually check the correctness of interactions encoded by the model.

Let us consider the explanation on the first input image as an example. We can analyze the representation quality of the DNN from the following three perspectives. (1) The interactions $I_{\text{AND}}(S = \{C, I\})$, $I_{\text{AND}}(S = \{D, I\})$, $I_{\text{AND}}(S = \{F, I\})$ and $I_{\text{AND}}(S = \{A, I\})$ between pedestrian patches and background patches may represent unreliable inference patterns. (2) High-order interactions, *e.g.*, $I_{\text{OR}}(S = \{A, C, D, E, F, G, H, I\})$ and $I_{\text{OR}}(S = \{A, C, D, F, H, I, J\})$, usually represent too complex inference patterns. Complex interactions usually have lower generalization power than simple interactions. (3) There are 29 positive interactions and 31 negative interactions extracted from an input image. The offsetting of positive and negative interactions is another problem. Adversarially robust neural networks usually encode more positive interactions and fewer negative interactions than normal neural networks.

In addition, the problematic interactions (*e.g.*, interactions on background patches) reflect representation flaws of a DNN, because it is found by (26) that salient interactions are usually transferable across different samples. In other words, problematic interactions may affect the inference of a large number of samples.

