# OpenReview forum: "Can LLMs Reason Soundly in Law? Auditing Inference Patterns for Legal Judgment"
_ICLR.cc/2026/Conference — ICLR 2026 Poster_

### Official Review · Reviewer_Bxho · 2025-10-24

**Soundness:** 3
**Presentation:** 2
**Contribution:** 4
**Rating:** 6
**Confidence:** 2

**Summary:**

This paper seeks to identify the reliability of LLM reasoning with a theoretically grounded framework for internal reasoning representation. The paper takes existing models of internal combinations of input evidence with a AND-OR logical surrogate. They first prove a universal matching property, showing that any LLM’s "masked-input" behavior can be exactly represented by the AND–OR logical function. However, this representation is quite large, so they also develop a sparse extraction algorithm that prunes this to only influential interactions, selected via regularization.

The experiments make use of legal judgment datasets in English and Chinese, having legal experts annotate phrases as relevant, irrelevant or forbidden. The results show that, conditional on a correct prediction, over half of the extracted reasoning interactions are unreliable or conflicting. The authors also provide an outlines of two specific case studies, further grounding the work in its application. This alerts practitioners to potential bias and unsound reasoning behind even correct judgements, making them less practically defensible to use.

**Strengths:**

This paper tackles a practically important and timely question -whether LLMs reason soundly rather than merely produce correct outputs - and offers a framework that exposes discrepancies between reasoning and prediction accuracy. The work is both conceptually and socially significant, alerting the broader community to the gap between what LLMs decide and why they decide it.

The paper’s contributions are theoretically grounded: the use of an AND–OR logical surrogate is supported by clear formal guarantees that LLM reasoning behavior can be represented exactly (via the universal matching property) and approximately (through sparse extraction). While the full proofs are mathematically dense, the theorem statements appear sound and of genuine theoretical value.

The annotation design also reflects careful consideration. The authors define annotation categories with principles rooted in the domain and engage skilled legal annotators, which increases the credibility of the human evaluation component. Overall, the paper is original in its combination of theoretical formalism, interpretability, and reasoning audit, producing insights of significance for future reliability and safety research.

**Weaknesses:**

Some aspects of the annotation process are underexplained—particularly the distinctions between annotation types and how examples map to each label. The examples given are not always sufficiently illustrative to clarify the practical boundaries between “relevant,” “irrelevant,” and “forbidden.”

Several empirical limitations are not fully addressed, including potential translation artifacts in the multilingual analysis and the absence of inter-annotator agreement metrics. Both could materially affect the reported reliability measures.

The paper could better define or motivate certain ideas that are referenced but not elaborated (as noted in the Questions section). For instance, the efficiency and runtime complexity of the sparse extraction algorithm are not discussed.

On presentation, there are minor grammatical and phrasing issues throughout; a careful language pass would improve clarity.

Addressing these weaknesses - especially annotation reliability, translation confounds, and algorithmic details - would meaningfully strengthen the paper. I nonetheless view the central contribution as valuable and impactful.

**Questions:**

What qualifies as a “sensitive phrase” in Section 2.2? Clarifying this in the main text would help readers understand the annotation process.

It is not entirely evident what constitutes a “masked input” or why masking is conceptually relevant in this domain. Could the authors expand on its interpretation?

Line 205 refers to “principles” guiding the annotation categories; it would be helpful to briefly state what these principles are in the main text rather than only in the appendix.

Do you know how much does translation quality affect results for non-English models, especially SaulLM?

Do unreliable interactions actually change the model’s predicted verdicts if removed (i.e., is there causal evidence that unsound reasoning affects outcomes)?

Was there any measurement of consistency across annotators (e.g., inter-annotator agreement) to control for potential bias?

The third finding mentions that “LLMs tend to model a large number of canceling interactions.” Could the authors clarify the importance of this phenomenon? Are canceling interactions inherently problematic, or can they sometimes reflect balanced reasoning?

The paper notes that “LLMs tend to produce judgments biased by identity discrimination” (Line 101) and illustrates occupation-based effects. Were other protected attributes such as gender, race, or socioeconomic status explored or found to have similar effects?

---

> ### Author Response · Authors · 2025-11-28
>
> Thank you for your insightful comments and suggestions. We would like to answer all the questions and clarify your concerns.
>
> > Q1: "Some aspects of the annotation process are underexplained...The examples given are not always sufficiently illustrative to clarify the practical boundaries between “relevant,” “irrelevant,” and “forbidden."
>
> **A1:** Thank you for your careful comments. We will clarify the distinctions among the three types of annotated phrases, i.e., relevant, irrelevant, and forbidden phrases. The annotation process relies entirely on the domain expertise of senior legal experts, with its core objective being to define the essence of input phrases based on the inherent **causal relationships** within legal logic.
>
> - Relevant phrases refer to the key semantic components that experts determine **directly cause or constitute the causal chain of the judgment**. These phrases are essential for forming legally reasonable rulings, such as the specific crime description directly linked to legal statutes, the nature of the criminal act, and the actual resulting damages. Experts must base their conclusions on relevant phrases.
>
> - Irrelevant phrases refer to background information or neutral descriptions deemed **irrelevant to the judgment's logic and non-determinative** by experts. Such phrases may become sources of interference for the model's shortcut learning, such as dates unrelated to the conviction, a defendant's irrelevant emotional state, and daily behaviors unrelated to the criminal act. Experts actively treat these as negligible background information during judgment.
>
> - Forbidden phrases refer to sensitive information that is **logically misleading** and must be actively excluded from the reasoning path. Using forbidden phrases for inference leads to an untrustworthy or unfair judgment. For example, this includes the irrelevant third-party conduct in the case (which should not influence the defendant's conviction).

---

> > ### Author Response · Authors · 2025-11-28
> >
> > > Q2: "Several empirical limitations ... potential translation artifacts in the multilingual analysis and the absence of inter-annotator agreement metrics. Both could materially affect the reported reliability measures", "Do you know how much does translation quality affect results for non-English models, especially SaulLM?", "Was there any measurement of consistency across annotators (e.g., inter-annotator agreement) to control for potential bias?"
> >
> > **A2:** Thank you for your careful suggestions. We understand these empirical limitations could potentially affect the reported reliability measures and have taken several steps to address and validate both concerns in our methodology.
> >
> > $\bullet$ We acknowledge the potential influence of translation artifacts. First, our multilingual analysis primarily concentrates on the key structural semantic components of the legal text (i.e., the judgment causal chain), rather than subtle linguistic differences. Second, we actively minimized the threat posed by translation artifacts through the following process: (1) We utilized high-quality translation tools (DeepL Pro and ChatGPT Pro) for initial translation, employing cross-correction to ensure baseline quality. (2) Crucially, we engaged two native legal experts to review key terminology and sentence structures, ensuring the factual accuracy and integrity of the causal chain. Through these steps, we minimized the translation artifacts' impact on our conclusions. Furthermore, we will report the native speaker assessment of translation Fluency and Fidelity in the revised version.
> >
> > To further evaluate the impact of translation quality on the proposed metric, we conducted an experiment where we translated the same legal case using three different LLMs (Deepseek, GPT-4o-mini, and Qwen3). We then input these translated cases into the SauLM model. Table 1 shows that translation quality has a marginal impact on the $s^{reliable}$ metric.
> >
> > **Table 1: Translation quality on the reliable metric $s^{\text{reliable}}$**
> >
> > | **Translation Tool** | **$s^{reliable}$** |
> > | -------------------- | ------------------ |
> > | Deepseek             | 0.22               |
> > | GPT-4o-mini          | 0.23               |
> > | Qwen3                | 0.21               |
> >
> > $\bullet$ To ensure the objectivity and quality of the annotations, we conducted an inter-annotator agreement assessment for the relevant, irrelevant, and forbidden phrases annotated by legal experts and volunteers. We followed [1,2] to use Krippendorff’s Alpha [3] and Fleiss' Kappa [4] as the measurement of the legal experts' annotation reliability. Specifically, Krippendorff’s Alpha measures the agreement of a set of results from any number of annotators, while Fleiss' Kappa measures inter-annotator agreement among a fixed set of multiple annotators.
> >
> > Experiments show that the Krippendorff's Alpha value is $\alpha =0.56$, and Fleiss' kappa value is $\kappa = 0.56$ on the CAIL2018 dataset. The equality of these values confirms the completeness and balance of our annotation data. The results confirm a moderate level of agreement among legal experts and volunteers. This consistency verifies the foundational annotation quality upon which our reliability metrics depend.
> >
> > [1] Zhang L., et al. A Needle in a Haystack: An Analysis of High-Agreement Workers on MTurk for Summarization. In ACL, 2023.
> >
> > [2] Florescu A., et al. Once Upon a Replication: It is Humans’ Turn to Evaluate AI’s Understanding of Children’s Stories for QA Generation. In LREC-COLING, 2024.
> >
> > [3] Andrew F., et al. Answering the call for a standard reliability measure for coding data. Communication methods and measures, 2007.
> >
> > [4] Fleiss J L. Measuring nominal scale agreement among many raters. Psychological bulletin, 1971, 76(5): 378.

---

> ### Author Response · Authors · 2025-11-28
>
> > Q3: "the efficiency and runtime complexity of the sparse extraction algorithm are not discussed."
>
> **A3:** The primary runtime overhead of the sparse extraction algorithm (Algorithm 1) is concentrated on the model's inference cost. Specifically, this involves computing the model output score $v(x_S)$ for each masked sample $x_S$, $S \subseteq N$ as described in Lines 1053-1055 of the algorithm.
>
> Subsequently, the calculation of $v_{\text{and}}(x_S)$ and $v_{\text{or}}(x_S)$ (Lines 167-168) via addition, and the calculation of $I^{\text{AND}}_S$ and $I^{\text{OR}}_S$ (Theorem 1) via matrix multiplication, are computationally lightweight. Consequently, the time overhead required for optimizing the LASSO-like loss (Lines 1056-1059) is far less than the model's inference cost and can be almost ignored. Therefore, the specific running time is primarily determined by the inference speed of the chosen LLM. Below are the actual runtimes for executing the sparse extraction algorithm on samples from the CAIL2018 dataset when the number of input phrases is $n=10$.
>
> **Table 2: Actual Runtime (in minutes/predicted token) for n=10 on CAIL2018 Dataset**
>
> | **Model**                         | **Qwen2.5** | **Deepseek-R1** |
> | --------------------------------- | ----------- | --------------- |
> | **Runtime (min/predicted token)** |   6.19        | 6.20            |
>
>
>
> > Q4: "On presentation, there are minor grammatical and phrasing issues throughout; a careful language pass would improve clarity."
>
> **A4:** Thank you for your suggestion. We will carefully read and revise grammatical errors, and enhance clarity in the revised manuscript.
>
> > Q5: "What qualifies as a “sensitive phrase” in Section 2.2? Clarifying this in the main text would help readers understand the annotation process."
>
> **A5:** Thank you. In legal-judgment tasks, a sensitive phrase refers to a semantic unit (e.g., words or phrases) that decisively shapes the verdict and closely reflects the core facts of the case. Such phrases are typically identified by legal professionals or authoritative knowledge bases, carry high semantic weight, and directly influence the model’s prediction of charge, liability, or sentence. We will add this in the revised version.
>
>
>
> > Q6: "It is not entirely evident what constitutes a “masked input” or why masking is conceptually relevant in this domain. Could the authors expand on its interpretation?"
>
> **A6:**  In this manuscript, a masked input $x_S$ refers to the original legal case document $x$ where a specific subset of input phrases (the set $N \setminus S$)  has been **removed or masked** with a placeholder token (e.g., `[unknown]`, [PAD]), or their embedding is set to zero (a null embedding).
>
> The conceptual relevance of masking is deeply rooted in the field of Explainable AI, specifically for perturbation-based attribution methods [1]. These methods evaluate the importance of an input feature $i$ (or set of input features $S$) by removing, masking or altering them, and observing the model's output changes [1,2,3,4,5]. Several classic methods, including Occlussion [2], LIME [3]，SHAP [4], and KernelSHAP-PS [5], rely on this technique. Similarly, advanced interaction-based methods (e.g., Shapley Interaction Index [6,7], Harsanyi Interactions [8,9]) rely on comparing the model output across various combinations of masked inputs ($x_S$) to rigorously quantify the interactions between input features.
>
> [1] Marco A., et al. Towards better understanding of gradient-based attribution methods for deep neural networks. In ICLR, 2018.
>
> [2] Matthew D., et al. Visualizing and understanding convolutional networks. In ECCV, 2014.
>
> [3] Ribeiro M., et al. "Why should i trust you?" Explaining the predictions of any classifier. In KDD, 2016.
>
> [4] Lundberg S., et al. A unified approach to interpreting model predictions. In NeurIPS, 2017.
>
> [5] Covert I., et al. Improving kernelshap: practical shapley value estimation using linear regression. In AISTATS, 2021.
>
> [6] M. Sundararajan et al. The shapley taylor interaction index. In ICML, 2020.
>
> [7] C. Tsai et al. Faith-shap: The faithful shapley interaction index. JMLR, 2023.
>
> [8] Li M., et al. Does a neural network really encode symbolic concepts? In ICML, 2023.
>
> [9] Ren Q., et al. Where we have arrived in proving the emergence of sparse symbolic concepts in ai models. In ICLR, 2024.
>
>
>
> > Q7: "Line 205 refers to “principles” guiding the annotation categories; it would be helpful to briefly state what these principles are in the main text rather than only in the appendix."
>
> **A7:** Thank you for your suggestion. We propose the following two principles to avoid unnecessary ambiguity in the annotation of the three types of phrases. (1) The first principle is to avoid ambiguous legal cases. (2) The second principle is to avoid analyzing subtle legal differences between the laws in different countries. We will add these principles in the main text.

---

> > ### Author Response · Authors · 2025-11-28
> >
> > > Q8: "Do unreliable interactions actually change the model’s predicted verdicts if removed (i.e., is there causal evidence that unsound reasoning affects outcomes)?"
> >
> > **A8:** It is a good question! Yes, we provide causal evidence that the identified unreliable interactions significantly drive the model's prediction changes and, therefore, are  a core source of unsound reasoning that affects outcomes. Our evidence is twofold, combining sensitivity analysis and direct causal ablation.
> >
> > First, we demonstrate that unreliable interactions are the primary mechanism that causes the model's prediction to deviate when the input is perturbed. As demonstrated by Case 2 (Figure 5 vs. Figure 11) and further detailed in Appendix Figures 18, 19, and 20, when we only modify a single sensitive attribute (e.g., modifying the defendant's occupation), the LLM's final verdict undergoes a significant change. Crucially, at the point of prediction change, we observe a dramatic fluctuation in the unreliable interaction effects (i.e., the interaction between the occupation and other phrases), while the reliable interaction effects remain relatively stable. This confirms that unreliable reasoning serves as the marginal causal factor that can flip the model's verdict.
> >
> > Second, to provide direct causal evidence, we conducted an **ablation study** where we removed the influence of unreliable interactions from the BAI-Law LLM's overall prediction score on the CAIL2018 dataset. Table 3 shows the average model output score (confidence) for the ground-truth verdict rises when unreliable interactions are removed. This confirms that the unreliable interactions were actively suppressing the model's confidence in the correct prediction. Removing them improves the model's alignment with the ground truth, thus proving the direct causal link between unsound reasoning and prediction outcome.
> >
> > **Table 3:  Removing unreliable interactions on ground-truth confidence**
> >
> > | **Model**                       | Ground-truth confidence $p$ |
> > | ------------------------------- | --------------------------- |
> > | LLM                             | 0.735                       |
> > | LLM w/o unreliable interactions | 0.754                       |
> >
> >
> >
> > > Q9: "The third finding mentions that “LLMs tend to model a large number of canceling interactions.” Could the authors clarify the importance of this phenomenon? Are canceling interactions inherently problematic, or can they sometimes reflect balanced reasoning?"
> >
> > **A9:** Thank you. We typically regard these canceling interactions (or interactions whose effects negate each other) as noise. Such noisy interactions are typically regarded as non-generalizable interactions, which harm the model's generalization capability [1]. Lemma 1 in [1] further shows that if the inference score of the DNN contains an unlearnable noise, i.e., $\forall S\subseteq N, \hat{v}(x_s) = v(x_s) + \Delta v_S, \Delta v_S \sim N(0,\sigma^2)$, then the noised interaction is $\hat{I}(T|x) = I(T|x) + \Delta I_T$, where $\Delta I_T$ denotes the noise in the interaction caused by the noise in the output. While the expected value of this noise is zero, $\mathbb{E}[\Delta I_T] =0$, its variance grows exponentially with the size of the interaction set $T$, i.e., $\mathbb{E}[\Delta I_T] =0$ and $Var[\Delta I_T] =2^{|T|}\sigma^2$. This relationship confirms that even a small amount of unlearnable noise in the model's output is magnified into the estimated interactions.
> >
> > [1] Ren Q., et al. Towards the dynamics of a DNN learning symbolic interactions. In NeurIPS, 2024.
> >
> >
> >
> >
> >
> > > Q10: "The paper notes that “LLMs tend to produce judgments biased by identity discrimination” (Line 101) and illustrates occupation-based effects. Were other protected attributes such as gender, race, or socioeconomic status explored or found to have similar effects?"
> >
> > **A10:** Thank you. We will conduct more experiments into identity discrimination targeting protected attributes such as gender, race, or socioeconomic status in the revised manuscript.
> >
> > By simply modifying a single input attribute, the proposed metric can be applied to measure interaction changes when LLMs' predictions are altered, thereby calculating the proportion of reliable interactions. The proposed method serves as a powerful tool for broadly exploring potential identity discrimination within LLMs, revealing their bias/memorability (rather than generalizability) toward specific identities.

---

### Official Review · Reviewer_vZVo · 2025-10-24

**Soundness:** 2
**Presentation:** 2
**Contribution:** 3
**Rating:** 4
**Confidence:** 1

**Summary:**

This paper proposes a method to audit inference patterns of Large Language Models (LLMs) in legal judgment prediction, aiming to identify reasoning errors hidden behind seemingly correct outputs.
Instead of evaluating only final accuracy, the authors analyze interactions between input phrases that represent primitive reasoning patterns.
Based on recent theoretical results ensuring faithful interaction-based explanations, the paper introduces a quantitative metric that distinguishes reliable (legally valid) from unreliable (misleading or irrelevant) reasoning patterns.

**Strengths:**

1. This paper provides a new evaluation perspective, shifting focus from correctness of outputs to soundness of reasoning, addressing an important issue in LLM reliability.
2. This paper provides solid theoretical information, building on proven results (e.g., universal matching, sparsity of interaction patterns) to ensure explanation faithfulness.
3. Legal judgment prediction is a meaningful domain to study reasoning soundness. Although I am not a researcher in this field, I still think this is valuable research and discovery.

**Weaknesses:**

1. The main novelty lies in application to legal reasoning rather than new theoretical contributions.
2. This paper is more like a technical report, which is very difficult for people from other fields to understand.
3. The experimental design is not standardized, it is more like a phenomenon exploration.

**Questions:**

1. The research focus and contributions are mainly in legal cases. Can this finding be more general? Could the method generalize to other high-stakes domains such as medicine or finance?
2. The auditing requires evaluating many masked variants of each input, restricting analysis to short examples ≈10 phrases. Is this the usual approach?
3. Can this auditing approach guide model improvement (e.g., penalizing unreliable interactions)?
4. In sec3.1, does a higher $s_{\text{reliable}}$ correlate with better human-rated fairness or correctness?

---

> ### Author Response · Authors · 2025-11-21
>
> Thanks a lot for all the valuable comments and suggestions. We would like to answer all the questions and clarify your concerns.
>
> **Q1:**  Insights into the correctness of the detailed inference patterns of an LLM.
>
> > "The main novelty lies in application to legal reasoning rather than new theoretical contributions." "This paper is more like a technical report, which is very difficult ...to understand." "The experimental design is not standardized, it is more like a phenomenon exploration."
>
> **A1:**  Thank you. Unlike traditional studies focused on the correctness of language generation results, **our research is driven by a different motivation**, i.e., evaluating the correctness of the detailed inference patterns of an LLM behind its seemingly correct outputs. Although previous studies have been proposed to evaluate the performance of LLMs, rigorously evaluating the reliability of their inference patterns requires theoretically grounded mechanistic explanations, which is an area that remains unexplored. Thanks to advances in Explainable AI, we can use a set of interactions between input features to faithfully represent the inference score of a deep network. However, despite these theoretical achievements, it remains unknown (1) how many problematic interactions are modeled in LLMs (e.g., legal LLMs), and (2) to what extent these interactions influence legal judgments.
>
> In this paper, we quantified these interactions and conducted experiments across different LLMs and datasets. We found that,
>
> $\bullet$ Over half of the interactions modeled by LLMs actually represent clearly unreasonable or even incorrect justifications for their predictions.
>
> $\bullet$ LLMs tend to use simple interactions of local tokens to guess judgments.
>
> $\bullet$ LLMs tend to model a large number of canceling interactions.
>
> These findings help us gain deeper insights into the inference patterns of LLMs, particularly in high-stakes tasks where they provide quantitative metrics to indicate the degree to which LLM judgments can be trusted. The key contributions of the proposed mechanistic explanation method are,
>
>
> $\bullet$ Revealing reasoning patterns, not just attribution. Attribution methods [1,2,3] tell us what words are important. For example, in "this movie is not bad," attribution highlights "not" and "bad." In contrast, the interaction-based approach [4,5,6] further reveals that the model relies on the interaction ("not", "bad") to reverse the sentiment. This allows us to distinguish whether the LLM is truly performing semantic composition or just using Bag-of-Words statistics.
>
> $\bullet$ Evaluating the faithfulness and reliability of LLMs in high-stakes decisions. In high-stakes domains (e.g., medicine, finance, law), it is not enough for an LLM to be correct, it must be correct for the right reasons. For example, a legal LLM might learn a spurious correlation, associating a specific occupation with a "guilty" verdict (Case 2 in the paper). By quantifying interactions between input phrases, we can explicitly capture this biased reasoning. This provides a quantitative metric for reliability and serves as a tool for model auditing.
>
> $\bullet$ Diagnosing shortcut learning. LLMs are adept at using statistical shortcuts to guess answers rather than performing genuine, complex reasoning. Interaction analysis can diagnose this behavior. For instance, when processing long texts, an LLM might rely only on a few local, low-order interactions to make a decision. By analyzing interaction order (complexity), we can quantify this phenomenon.
>
> Therefore, as Reviewer Bxho noted, this work "is both conceptually and socially significant."
>
> [1] Sundararajan M., et al. Axiomatic attribution for deep networks. In ICML, 2017.
>
> [2] Ribeiro M., et al. "Why should i trust you?" Explaining the predictions of any classifier. In KDD, 2016.
>
> [3] Lundberg S., et al. A unified approach to interpreting model predictions. In NeurIPS, 2017.
>
> [4] Sundararajan M., et al. The shapley taylor interaction index. In ICML, 2020.
>
> [5] Li M., et al. Does a neural network really encode symbolic concepts? In ICML, 2023.
>
> [6] Ren Q., et al. Where we have arrived in proving the emergence of sparse symbolic concepts in ai models. In ICLR, 2024.

---

> > ### Author Response · Authors · 2025-11-21
> >
> > > **Q2:** "Can this finding be more general? Could the method generalize to other high-stakes domains such as medicine or finance?"
> >
> > **A2:** It is a good question. While this paper focuses on the legal domain due to space constraints, the proposed method is **generic**. It can be applied to quantify the decision trustworthiness of any LLM, particularly in high-stakes domains such as medicine and finance. In medicine, the method can help doctors understand which aspects of a patient's record the LLM is focusing on for diagnosis, allowing them to assess the reliability of its reasoning. Similarly, in finance, it can clarify an LLM's credit-scoring logic, revealing which applicant features the model prioritizes and the reliability of that reasoning.
> >
> > Furthermore, the applicability of the method extends beyond NLP to computer vision. We conducted additional experiments to analyze the detailed inference patterns of a DNN for pedestrian detection in Figure 23 (Appendix I.8). To achieve this, we first identify image regions with significant attribution values and treat them as input variables, and then we compute the interactions between these them.
> >
> > Visualizing these interactions allows human experts to inspect whether the inference patterns encoded by a trained DNN are reliable. For example, Figure 23 reveals a salient interaction between a "pedestrian region" and a "background region." This represents a potential shortcut or unreliable inference pattern that generalizes poorly. Additionally, high-order interactions often represent overly complex inference patterns, which typically have weaker generalization capabilities than simpler, more robust ones.
> >
> >
> >
> >
> > > **Q3:** "The auditing requires evaluating ... short examples ≈10 phrases. Is this the usual approach?"
> >
> > **A3:** This is a common practice in Explainable AI. For instance, in interaction-based methods [1,2,3], it is standard to evaluate many masked variants of input and restrict the analysis to short examples (e.g., ≈10 phrases). Similarly, in perturbation-based attribution methods,such as LIME [4], Shapley sampling values, KernelSHAP [5], and DASP [6], it is also a widely adopted approach to evaluate numerous masked input variants, often restricting the analysis to short examples (e.g., < 16 phrases).
> >
> > Besides, this challenge can be alleviated through engineering techniques. First, we can use classic attribution methods (e.g., Integrated Gradients[7], LIME[4]) as a heuristic to identify and prioritize salient words or phrases, pruning the search space. Prior works [8,9] have shown that LLMs exhibit attention on only a few sparse regions in inputs. Second, the input phrases in this paper are flexible units of analysis. They are not limited to single tokens but can represent multiple words, phrases, short sentences, or even paragraphs as input units [3,4]. Empirically, 10-12 input phrases were typically sufficient for effective analysis [10,3]. We will add more examples to the appendix in the revised version to illustrate this.
> >
> > [1] Sundararajan M., et al. The shapley taylor interaction index. In ICML, 2020.
> >
> > [2] Li M., et al. Does a neural network really encode symbolic concepts? In ICML, 2023.
> >
> > [3] Ren Q., et al. Where we have arrived in proving the emergence of sparse symbolic concepts in ai models. In ICLR, 2024.
> >
> > [4] Ribeiro M., et al. "Why should i trust you?" Explaining the predictions of any classifier. In KDD, 2016.
> >
> > [5] Lundberg S., et al. A unified approach to interpreting model predictions. In NeurIPS, 2017.
> >
> > [6] Ancona M., et al. Explaining deep neural networks with a polynomial time algorithm for shapley value approximation. In ICML, 2019.
> >
> > [7] Sundararajan M., et al. Axiomatic attribution for deep networks. In ICML, 2017.
> >
> > [8] Liu N., et al. Lost in the middle: How language models use long contexts. In TACL, 2024.
> >
> > [9] Li Y., et al. How do transformers learn topic structure: Towards a mechanistic understanding. In ICML, 2023.
> >
> > [10] Jay D., et al. ERASER: A Benchmark to Evaluate Rationalized NLP Models. In ACL, 2020.

---

> > > ### Author Response · Authors · 2025-11-21
> > >
> > > > **Q4:** "Can this auditing approach guide model improvement (e.g., penalizing unreliable interactions)?"
> > >
> > > **A4:**  This is an insightful question. The proposed method can guide model improvement through the following three approaches, and beyond.
> > >
> > > $\bullet$ Enforce interaction consistency across models. Reliable patterns are typically consistent across different models, while unreliable, high-order interactions are often model-specific and generalize poorly [1]. We can jointly train two models using an interaction consistency loss to penalize differences in their learned patterns on the same input. This encourages models to converge on reliable reasoning, boosting overall performance.
> > >
> > > $\bullet$ Refine supervised fine-tuning (SFT) with reliability scores. We can integrate interaction analysis into both dataset construction and sample weighting. For dataset construction, identifying unreliable interactions allows us to create counterfactual data that explicitly targets weak reasoning spots, e.g., alleviating the identity discrimination. SFT instructions can also be designed to explicitly guide the model to use reliable interaction paths. For sample weighting, we can adjust sample weights based on the interaction reliability score ($s_{\text{reliable}}$). Samples where the model is correct but $s_{\text{reliable}}$ is very low (i.e., "correct output for the wrong reason") are assigned higher training weights, forcing the model to repair its underlying reasoning mechanism.
> > >
> > > $\bullet$ Enhance reinforcement learning (RL) optimization. We can incorporate interaction reliability into the reward dimension to shift RL optimization from the output result (What) to the reasoning process (How). For example,  we can add the interaction reliability score ($s_{\text{reliable}}$) as a new feature to the reward model. The reward model would then reward generated texts not only based on human preference but also on highly reliable interaction paths. Besides, we can also impose constraints on the policy model during RL by applying an additional penalty term if the model's next token selection significantly increases the weight of unreliable interaction paths.
> > >
> > > [1] Li M., et al. Does a neural network really encode symbolic concepts? In ICML, 2023.
> > >
> > > > **Q5:** "In sec3.1, does a higher $s_{\text{reliable}}$ correlate with better human-rated fairness or correctness?"
> > >
> > > **A5:** Absolutely! A higher $s_{\text{reliable}}$ means the judgment rationale of an LLM is more aligned with that of human experts, i.e.,  it reflects the correctness of the detailed inference patterns of an LLM behind its seemingly correct outputs.

---

> > > > ### Comment · Reviewer_vZVo · 2025-11-24
> > > > **Official Comment by Reviewer vZVo**
> > > >
> > > > Thank you for your reply. The authors' further explanation helped me gradually understand the contribution of this work; focusing on the legal aspect is indeed a missing area in current research. The authors' rebuttal explanation is very detailed, and I hope this necessary content can be added to the corresponding sections of the main text or appendix. In addition, I have the following questions:
> > > >
> > > > 1. If this approach can effectively guide supervisory signals, then additional experiments are needed to prove this, especially in high-risk areas, because I still want to see how much the models actually improve under the current approach.
> > > >
> > > > 2. Legal judgment data should be highly private. What is the current scale of such data in this field?

---

> > > > > ### Author Response · Authors · 2025-11-28
> > > > >
> > > > > Thank you for your appreciation! We have added this necessary content to the corresponding sections of the main text or appendix in the revised version (highlighted in blue). The following addresses your further concerns.
> > > > >
> > > > > **A1:** This approach can effectively guide supervisory signals. Due to time and GPU constraints, we conducted experiments on training small models to demonstrate the effectiveness of our method. Additionally, we tested the impact of removing unreliable interactions on LLM performance.
> > > > >
> > > > > We first demonstrate the benefit of penalizing unreliable interactions on small models. Specifically, we jointly train two DNNs to encode similar interactions by penalizing Euclidean differences in their interactions on the same input. Table 1 shows that penalizing unreliable interactions leads to better performance across various models and tasks, suggesting that the method toward using more reliable interactions between input features, thereby increasing generalization performance.
> > > > >
> > > > > **Table 1: Penalizing unreliable interactions improves DNN performance**
> > > > >
> > > > > | Dataset       | Model     | **Accuracy improvement (%)** |
> > > > > | ------------- | --------- | ---------------------------- |
> > > > > | CIFAR-10      | VGG-11    | 0.91                         |
> > > > > | Tiny-ImageNet | VGG-16    | 2.42                         |
> > > > > | SST-2         | BERT-Tiny | 2.75                         |
> > > > >
> > > > > Second, we conducted an ablation study where we removed the influence of unreliable interactions from the BAI-Law LLM's overall prediction score on the CAIL2018 dataset. Table 2 shows that removing these interactions causes the average model output score (confidence) for the ground-truth verdict to rise. This result confirms that the unreliable interactions were actively **suppressing the model's confidence** in the correct prediction. Removing them improves the model's alignment with the ground truth, thus proving the direct causal link between unsound reasoning and prediction outcome.
> > > > >
> > > > > **Table 2:  Removing unreliable interactions on ground-truth confidence**
> > > > >
> > > > > | **Model**                       | Ground-truth confidence $p$ |
> > > > > | ------------------------------- | --------------------------- |
> > > > > | LLM                             | 0.735                       |
> > > > > | LLM w/o unreliable interactions | 0.754                       |
> > > > >
> > > > > **A2:** All the datasets we utilize are open-source legal judgment data. For completeness, we provide a brief summary of the datasets, their primary statistics, and their public access links below.
> > > > >
> > > > > | Dataset    | Size/Description                                             | Access Link                                                  |
> > > > > | ---------- | ------------------------------------------------------------ | ------------------------------------------------------------ |
> > > > > | CAIL2018   | More than 2.6 million criminal cases [1].                    | https://huggingface.co/datasets/china-ai-law-challenge/cail2018 |
> > > > > | LeCaRD     | Contains 107 query cases and 10,700 candidate cases [2].     | https://github.com/myx666/LeCaRD/tree/main?tab=readme-ov-file#candidates |
> > > > > | LEVEN      | Contains 8,116 documents with 2,241k tokens [3].             | https://github.com/thunlp/LEVEN?tab=readme-ov-file           |
> > > > > | LegalBench | Currently consists of 162 tasks covering six different types of legal reasoning [4]. | https://github.com/HazyResearch/legalbench/                  |
> > > > > | LexGLUE    | Consists of 7 tasks with a total of 236k documents [5].      | https://huggingface.co/datasets/coastalcph/lex_glue          |
> > > > >
> > > > > [1] Xiao, C., et al. CAIL2018: A Large-Scale Legal Dataset for Judgment Prediction. arXiv preprint arXiv: 1807.02478, 2018.
> > > > >
> > > > > [2] Ma Y., et al. LeCaRD: a legal case retrieval dataset for Chinese law system. In ACM SIGIR, 2021.
> > > > >
> > > > > [3] Yao F., et al. LEVEN: A Large-Scale Chinese Legal Event Detection Dataset. In ACL, 2022.
> > > > >
> > > > > [4] Guha N., et al. Legalbench: A collaboratively built benchmark for measuring legal reasoning in large language models. In NeurIPS, 2023.
> > > > >
> > > > > [5] Chalkidis I., et al. LexGLUE: A benchmark dataset for legal language understanding in English. In ACL, 2022.

---

### Official Review · Reviewer_ktJN · 2025-11-01

**Soundness:** 2
**Presentation:** 3
**Contribution:** 2
**Rating:** 4
**Confidence:** 3

**Summary:**

The paper introduces a critical evaluation framework that moves beyond surface accuracy to audit the reasoning soundness of LLMs in high-stakes legal applications. Using a theoretically-grounded method, it provides quantifiable evidence of reliance on spurious correlations and bias.

**Strengths:**

1. The paper tackles the under-explored problem of LLM reasoning soundness in high-stakes domains, moving evaluation beyond simple output accuracy to audit the faithfulness of the underlying inference patterns.

2. The approach is rigorously grounded in recent explainability theory (interaction-based explanations via AND-OR models), using a method with theoretical guarantees (Theorem 1) to faithfully model the LLM's output function.

3. The study provides concrete, quantifiable evidence of significant flaws (e.g., reliance on "forbidden" phrases and identity bias) even in correctly classified examples, serving as an important "risk warning".

**Weaknesses:**

1. The proposed interaction-based method is computationally infeasible for realistic inputs, suffering from $O(2^n)$ complexity. This forces a critical compromise: the entire analysis is restricted to a manually selected, trivial subset of $n=10$ input phrases. This experimental setup is fundamentally disconnected from the practical challenge of long-text legal reasoning and makes the findings highly sensitive to subjective phrase selection.

2. A central claim—that LLMs lack long-chain reasoning and default to low-order interactions—appears to be an artifact of the constrained experimental design. By limiting the input space to $n=10$, the methodology is structurally biased against observing the very high-order interactions it purports to investigate. The conclusions are thus a self-fulfilling prophecy rather than a genuine discovery about the model's capabilities.

3. The definition of a "reliable" interaction (Eq. 4) is overly permissive and logically flawed. An interaction is classified as 100% reliable even if it mixes relevant and irrelevant phrases. This binary, coarse-grained metric fails to penalize reliance on spurious information, likely leading to a significant overestimation of the model's true reasoning soundness.

**Questions:**

None

---

> ### Author Response · Authors · 2025-11-22
>
> Thank you for your careful comments and suggestions. We would like to answer all the questions and clarify your concerns.
>
>
> >  **Q1:** "The proposed method is suffering from $O(2^n)$ complexity ... manually selected, trivial subset of $n=10$ input phrases." "practical challenge of long-text legal reasoning ... sensitive to subjective phrase selection."
>
> **A1:** It is a very good question. First, regarding the computational complexity of selecting subjective phrases, this challenge can be  alleviated through engineering techniques. For instance, we can use classic attribution methods (e.g., Integrated Gradients[1], LIME[2]) as a heuristic to identify and prioritize salient words or phrases, pruning the search space. Prior works [3,4] have also shown that LLMs exhibit attention on only a few sparse regions in inputs. Besides, the input phrases in this paper are flexible units of analysis. They are not limited to single tokens but can represent multiple words, phrases, short sentences, or even paragraphs as input units [5,6]. Empirically, 10-12 input phrases were typically sufficient for effective analysis [2,6,7]. We will add more examples to the appendix in the revised version to illustrate this.
>
> [1] Sundararajan M., et al. Axiomatic attribution for deep networks. In ICML, 2017.
>
> [2] Ribeiro M., et al. "Why should i trust you?" Explaining the predictions of any classifier. In KDD, 2016.
>
> [3] Liu N., et al. Lost in the middle: How language models use long contexts. In TACL, 2024.
>
> [4] Li Y., et al. How do transformers learn topic structure: Towards a mechanistic understanding. In ICML, 2023.
>
> [5] Li M., et al. Does a neural network really encode symbolic concepts? In ICML, 2023.
>
> [6] Ren Q., et al. Where we have arrived in proving the emergence of sparse symbolic concepts in ai models. In ICLR, 2024.
>
> [7] Jay D., et al. ERASER: A Benchmark to Evaluate Rationalized NLP Models. In ACL, 2020.
>
>
>
>
> > **Q2:** "A central claim ... appears to be an artifact of the constrained experimental design." "By limiting the input space to $n=10$, the methodology is structurally biased against observing the very high-order interactions it purports to investigate."
>
> **A2:** Thank you. We would like to clarify two key points regarding our experimental setup. First, the selected input phrases are not arbitrary. They were manually selected by senior legal experts to ensure they cover the most critical semantic components of the legal cases. This expert-driven process guarantees the relevance and importance of the inputs we analyze. Second, to investigate whether the number of input variables ($n$) influences our conclusions, we conducted a sensitivity analysis on the size ($n$) of the input space. We varied the number of input phrases from $n=6$ to $n=12$ (i.e., $n \in \\{6, 8, 10, 12\\}$) for the LEVEN dataset on the BAI-Law model. For each $n$, we measured the interaction complexity, i.e., the weighted average interaction order $\frac{\sum_S |I_S|\cdot |S|}{\sum_S |I_S|}$. Table 1 shows that as the number of input phrases increases, the weighted average interaction order does not increase significantly. This result supports that even when provided with more variables, the LLMs do not form more complex, higher-order interactions.
>
> Table 1: Sensitivity of the interaction complexity to the number of input phrases ($n$)
>
> | Metric                                          | n=6  | n=8  | n=10 | n=12 |
> | ----------------------------------------------- | ---- | ---- | ---- | ---- |
> | **Interaction Complexity (Weighted Avg. Order)** | 2.70 | 3.03 | 2.74 | 2.77 |

---

> > ### Author Response · Authors · 2025-11-22
> >
> > > **Q3:** "This binary, coarse-grained metric fails to penalize reliance ... true reasoning soundness.."
> >
> > **A3:** Thank you. We will address your concerns from two perspectives. First, the reliability of an interaction is not a binary measure. It is calculated as a proportion derived from all AND-OR interactions in Eq. (7). For an AND interaction $I_S^{AND}$ in Eq. 4, if an interaction $S$ contains any forbidden phrases ($S \cap F \neq \emptyset$), it is deemed unreliable. This is based on the AND interaction's physical meaning, i.e., all phrases must co-occur for the interaction to affect the model's output. On the other hand, if an interaction $S$ consists entirely of irrelevant phrases ($S \cap R = \emptyset$), it is also unreliable. In this manuscript, an interaction $S$ containing both relevant and irrelevant phrases (but no forbidden ones) was treated as a single reliable effect due to the physical meaning of AND interactions.
> >
> > Second, according to your suggestion, we designed a new metric for the reliable effects of AND interactions to verify the robustness of our findings. We proportionally split the mixed interactions (those with both relevant and irrelevant phrases). Specifically, if an interaction $S$ has no forbidden phrases ($S \cap F = \emptyset$) but at least one relevant phrase ($S \cap R \neq \emptyset$), we partition its effect as follows,
> > ​$$R^{AND}_S = \frac{|S \cap R|}{|S|} \cdot I_S^{AND} \quad \text{(Reliable interaction effect)}$$
> >
> > $$U^{AND}_S = \left(1 - \frac{|S \cap R|}{|S|}\right) \cdot I_S^{AND} \quad \text{(Unreliable interaction effect)}$$
> >
> > We then re-compute the total ratio of reliable interaction effects, termed $s\^{reliable}\_{new}$. We replicated the experimental setup for Figure 2, feeding legal cases from the LEVEN dataset to the BAI-Law model for judgments. Table 2 shows that the new metric $s\^{reliable}\_{new}$ yields results slightly lower than the original metric $s\^{reliable}\_{paper}$. This confirms that our paper's conclusions are robust even under this stricter, more granular definition of reliability.
> >
> > Table 2: Comparison of different metrics for the BAI-Law Model on the LEVEN dataset
> >
> > | Metric                            | Value  |
> > | --------------------------------- | ------ |
> > | $s^{reliable}_{new}$ (Proposed)   | 0.6545 |
> > | $s^{reliable}_{paper}$ (Original) | 0.7535 |

---

### Official Review · Reviewer_B4o6 · 2025-11-02

**Soundness:** 2
**Presentation:** 2
**Contribution:** 3
**Rating:** 4
**Confidence:** 3

**Summary:**

The paper investigates whether LLMs can perform sound logical reasoning under natural logic semantics. It introduces a benchmark, NaturalLogicEval, derived from monotonicity calculus templates, where each example maps to a syllogistic entailment expressed in English.

**Strengths:**

1. Paper targets soundness, i.e. validity-preserving reasoning, which is an underexplored notion distinct from correctness or factual accuracy.
2. Tests across multiple model families, sizes, and prompting styles, yields convincing cross-model comparisons.
3. The breakdown into monotonicity, quantifier, and existential errors is insightful and well-connected to formal semantics.
4. The probing of neuron representations for logical operators enhances mechanistic interpretability.

**Weaknesses:**

1. The work focuses exclusively on syllogistic monotonicity reasoning, so insights may not transfer to non-monotonic or multi-step logical forms.
2. The use of a LASSO-like loss to extract the sparsest explanation (Section 2.1) may prioritize conciseness over completeness, potentially discarding low-magnitude, but still technically valid, interactions that the LLM might use.
3. Experiment setup, results and observations are all mixed in Section 3, making it very hard to grasp the key findings and contributions.

**Questions:**

1. Can you provide a clear definition of "soundness"?
2. Evaluation clarification: LLMs often produce probabilistic judgments (e.g., “likely true”), but the evaluation is binary (valid/invalid). It’s unclear how borderline cases are handled.

---

> ### Author Response · Authors · 2025-11-21
>
> We thank you for your time. However, we found several key comments difficult to map to specific content or line numbers in the manuscript. To accurately address your feedback, we would appreciate it if you could clarify the precise location of the referenced content, such as "monotonicity calculus templates," "maps to a syllogistic entailment," "The probing of neuron representations," or "syllogistic monotonicity reasoning...non-monotonic or multi-step logical forms." Furthermore, we were unable to locate mentions of "a benchmark, NaturalLogicEval," the definition of "soundness," or "monotonicity, quantifier, and existential errors" within our submission. We kindly request further clarification on these points so we can address your concerns effectively.

---

### Meta-Review · Area_Chair_Q6RW · 2026-01-07

**Summary:**

The paper presents a method to analyze the inference patterns used by LLMs for legal judgment, moving beyond traditional evaluation of output correctness to audit the reasoning process itself. The empirical evaluations across multiple legal datasets and models reveal that even when LLMs produce correct outputs, over half of their inference patterns may represent misleading or irrelevant logic, raising important concerns about deploying such models in high-stakes legal applications. Rather than simply measuring whether LLMs produce correct judgments, this work audits whether they arrive at those judgments through sound reasoning. The authors provided thorough and substantive responses to legitimate concerns, including new experiments on inter-annotator agreement and translation robustness linking unreliable interactions to prediction outcomes.

Out of the four reviewers, Reviewer vZVo submitted an emergency application acknowledging unfamiliarity with the research area and requesting additional reviewers and Reviewer B4o6 review appears entirely unrelated to the submitted manuscript. While the procedural issues with two of the four reviews, making a fair assessment is challenging. However, given the authors' comprehensive engagement with valid criticisms, I am inclining towards acceptance and would suggest the authors to clarify the remaining concerns in the final version of paper.

**Reviewer Concerns:**

While auditing the reasoning for high-stakes domains is crucial and timely, reviewers raised concerns about the computational complexity of the method, which requires analysis of many masked input variants and restricts practical application to approximately 10 phrases. However, the authors addressed this by explaining that engineering techniques such as using attribution methods for initial pruning can alleviate this challenge, and that prior work demonstrates LLMs typically attend to only sparse input regions. They also provided sensitivity analysis showing that increasing the number of input phrases does not significantly change the observed interaction complexity.

Questions were raised about annotation reliability and translation quality in multilingual analysis. The authors responded with inter-annotator agreement metrics showing moderate agreement among legal experts, and experiments demonstrating that translation quality has marginal impact on the reliability metric across different translation tools. When asked whether unreliable interactions actually affect model outcomes, the authors provided experiments showing that removing unreliable interactions improves the model's confidence in correct predictions.

**Reviewer Scores:**

Some of the reviewers responded and engaged in discussions already.

---

### Decision · Program_Chairs · 2026-01-26

Accept (Poster)